# Graph Diffusion that can Insert and Delete

**Matteo Ninniri**
Department of Computer Science
University of Pisa
56127 Pisa (Italy)
`matteo.ninniri@phd.unipi.it`

**Marco Podda**
Department of Computer Science
University of Pisa
56127 Pisa (Italy)
`marco.podda@unipi.it`

**Davide Bacciu**
Department of Computer Science
University of Pisa
56127 Pisa (Italy)
`davide.bacciu@unipi.it`

## Abstract

Generative models of graphs based on discrete Denoising Diffusion Probabilistic Models (DDPMs) offer a principled approach to molecular generation by systematically removing structural noise through iterative atom and bond adjustments. However, existing formulations are fundamentally limited by their inability to adapt the graph size (that is, the number of atoms) during the diffusion process, severely restricting their effectiveness in conditional generation scenarios such as property-driven molecular design, where the targeted property often correlates with the molecular size. In this paper, we reformulate the noising and denoising processes to support monotonic insertion and deletion of nodes. The resulting model, which we call GRIDDD, dynamically grows or shrinks the chemical graph during generation. GRIDDD matches or exceeds the performance of existing graph Diffusion Models on molecular property targeting despite being trained on a more difficult problem. Furthermore, when applied to molecular optimization, GRIDDD exhibits competitive performance compared to specialized optimization models. This work paves the way for size-adaptive molecular generation with graph diffusion.

## 1 Introduction

Generating molecules conditioned on predefined structures or properties is a central endeavor in computational chemistry, with applications ranging from *de novo* drug design [You et al., 2018a] to materials discovery [Zhao et al., 2023]. Depending on the conditioning information, we distinguish two key learning tasks: *property targeting*, aimed at generating molecules endowed with prespecified properties; and *property optimization*, where the goal is to edit a given molecule to improve a target property while retaining its core structure. Unlike continuous data, generating graphs must account for their discrete and combinatorial connectivity. For molecules in particular, chemical constraints invalidate most atom-bond and atom-atom combinations, making this problem difficult and, therefore, actively researched. Despite these challenges, deep graph generators [Faez et al., 2021] have achieved striking success in the above-mentioned tasks, due to their ability to approximate complex molecular distributions and their flexibility in incorporating conditioning information.

Based on the way they decode a graph from a latent representation, deep generative models for molecules can be broadly categorized as autoregressive or one-shot [Zhu et al., 2022]. Autoregressive models build the graph sequentially (atom-by-atom as in You et al. [2018b], or fragment-by-fragment

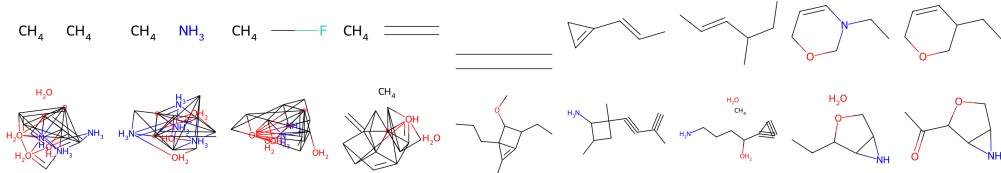

Figure 1: Two qualitative examples of the proposed model (GRIDDD) when generating molecules from the QM9 dataset. In the top row, from left to right, we show a subset of the denoising process to generate a molecule starting from a latent graph with two atoms (which are extremely rare in QM9). GRIDDD successfully inserts six more nodes to obtain a sample resembling the training set's distribution. In the bottom row, we start from a latent with 18 atoms instead (not present in QM9). GRIDDD manages to delete nine atoms and obtain a valid molecule. Notice that, unlike current DDPMs for graphs, the graph size is changed dynamically during denoising.

as in Jin et al. [2018b]), but suffer from order dependence and linearly-scaling sampling time. One-shot methods generate all nodes and edges in a single pass enabling parallel sampling, but struggle to learn high-order interactions. Denoising Diffusion Probabilistic Models (DDPMs) [Ho et al., 2020] for graphs bridge these approaches by progressively removing Gaussian or discrete noise to recover the molecular graph. Indeed, DDPMs preserve permutation invariance and sampling parallelism while iteratively refining long-range interactions (e.g., ring closures), which traditional one-shot decoders are forced to learn in a single transformation. Moreover, DDPMs can be easily adapted to conditional generation with classifier-based [Dhariwal and Nichol, 2021] or classifier-free [Ho and Salimans, 2022] guidance.

One drawback of Diffusion Models (not limited to graphs) is that the sample size remains fixed throughout the generative process. While for different modalities this is less of a concern (e.g., for images it is equivalent to fixing the resolution beforehand), it becomes relevant when generating molecules, and more in general when modeling combinatorial data. In practical terms, this restriction implies that the model *by design* cannot add or remove atoms during generation, leading to two major drawbacks: *a)* in property targeting, it is not possible to make the molecular structure responsive to properties which correlate with the size, e.g., when generating molecules with a specific molecular weight; *b)* in property optimization, the model cannot optimize a given property by adapting the generated structure. Existing methods circumvent this limitation by sampling the graph size before the generative process starts. For example, Vignac et al. [2023a] and Hoogeboom et al. [2022] sample different graph sizes from the empirical distribution of the training dataset, while Ninniri et al. [2024] use an auxiliary classifier to predict the graph size from the conditioning information. Though these workarounds are sufficient for property targeting, they become ineffective for property optimization, where size is strictly related to the structure to optimize. Ketata et al. [2025] adapts to property optimization by sampling additional nodes in the reverse process, but does not incorporate this procedure into the training process.

Motivated by these considerations, our primary contribution is a generalization of the standard discrete diffusion process on graphs. Our approach enables step-wise monotonic node insertions and removals during diffusion, and is specifically designed to change the graph size dynamically throughout the generative process, providing the necessary flexibility to incorporate conditional information more effectively. We implement this novel formulation as a model called GRIDDD (short for Graph Insert-Delete Discrete Diffusion). An intuitive example of how GRIDDD generates molecules is provided in Figure 1. We test GRIDDD on property targeting in two widely used benchmarks (QM9 and ZINC-250k), where it consistently performs on par or better than the state of the art in terms of approximating the target property while keeping high chemical validity, despite having been trained on a more difficult problem. When applied to molecular optimization, GRIDDD convincingly outperforms other molecular optimizers, achieving a higher average improvement and optimization success rate. To the best of our knowledge, this is the first work that allows size-adaptive molecular generation with graph diffusion. Our code is available at https://github.com/mninniri/GrIDDD.

## 2 Background and related works

**Notation**   We represent molecules with $n$ atoms as graphs $G = (\boldsymbol{X}, \boldsymbol{E})$, where $\boldsymbol{X} \in \mathbb{R}^{n \times a}$ is the matrix of one-hot encoded atom types (over $a$ possible atom types), and $\boldsymbol{E} \in \mathbb{R}^{n \times n \times b}$ is an adjacency (or edge) tensor that encodes both the bond connectivity and the one-hot encoded bond types (over $b$ possible bond types while considering the absence of a bond as an additional bond type). We denote as $\boldsymbol{x}_i \in \mathbb{R}^a$ the $i$-th row of $\boldsymbol{X}$ and as $\boldsymbol{e}_{ij} \in \mathbb{R}^b$ the slice of $\boldsymbol{E}$ along the first and second dimensions. Oftentimes, we use the terms "nodes" and "edges" to refer to atoms and bonds, respectively. Conditioning vectors will be generally denoted as $\boldsymbol{y} \in \mathbb{R}^d$, $d$ being a positive integer.

### 2.1   Graph Generative Models for Property Targeting and Optimization

Several models have recently been developed for property targeting, mostly based on graph diffusion since it is particularly suited for conditional generation. For example, DiGress [Vignac et al., 2023a] uses a categorical diffusion process on nodes and edges, and a graph transformer denoiser to learn a generative distribution over molecular graphs. DiGress allows conditional sampling through classifier-based guidance, by training an auxiliary regressor to steer the generation towards the target properties. FreeGress [Ninniri et al., 2024] builds upon DiGress and proposes a classifier-free approach for conditional generation that injects the guide directly into the denoiser during training, greatly improving property-targeting accuracy.

Molecular property optimization is mainly achieved through translation models and optimization models. VJTNNs [Jin et al., 2018b], Seq2Seq models [He et al., 2021], and HierG2G [Jin et al., 2020] all work by translating between different types of molecular representations, ranging from junction trees to SMILES strings. While these models achieve good results, they require a dataset of pairs of similar molecules with specific properties, which are of limited availability. At the same time, they are not designed to generate new molecules from scratch. Optimization models are usually designed for molecular generation. GCPN [You et al., 2018a] and JT-VAE [Jin et al., 2018a] employ different techniques, ranging from reinforcement learning to junction trees, to generate data, and are easily extensible to perform property targeting and optimization.

### 2.2   Discrete Denoising Diffusion Probabilistic Models

DDPMs consist of an un-parameterized *forward process* $q(\boldsymbol{x}^t | \boldsymbol{x}^{t-1})$ and a parameterized *reverse process* $p_\theta(\boldsymbol{x}^{t-1} | \boldsymbol{x}^t)$. The forward process progressively corrupts the initial data point $\boldsymbol{x}^0$, transforming it into a noise-like sample $\boldsymbol{x}^T$. The reverse process, trained to denoise these samples, aims to reconstruct $\boldsymbol{x}^0$ by sequentially removing the added noise. Sampling from a trained DDPM involves drawing $\tilde{\boldsymbol{x}}^T$ from the noise distribution and applying the reverse process iteratively over $T$ steps to obtain a new sample $\tilde{\boldsymbol{x}}^0 \sim q(\boldsymbol{x})$. While in the original formulation of DDPMs the noise distribution is Gaussian, we focus on *discrete* DDPMs where noise is injected through specially crafted *transition matrices* $\boldsymbol{Q}^t$, with $[\boldsymbol{Q}^t]_{ij} = p(x^t = j | x^{t-1} = i)$, encapsulating the probability of switching from category $i$ to category $j$ at step $t$ of the forward process. Crucially, cumulative transitions from an arbitrary timestep $s$ to $t$ can be expressed as a matrix $\overline{\boldsymbol{Q}}_{t|s} = \prod_{i=s+1}^{t} \boldsymbol{Q}^i$. The reverse process $p_\theta(\boldsymbol{x}^{t-1} | \boldsymbol{x}^t)$ is computed by marginalization over the possible categories of the input $\boldsymbol{x}$.

### 2.3   Insert and delete operations for Diffusion Models

The idea of insert and delete operations for discrete Diffusion Models originates from Johnson et al. [2021], which, similarly to us, proposed to gradually insert or delete nodes both during the forward and reverse process. However, there are some major differences with our work. Firstly, our solution is designed for graphs, instead of textual data. Secondly, they resort to complex edit summaries to compute the various probability distributions involved, while we simplify the computation of the posteriors by generalizing the denoising process to account for nodes that have yet to be deleted or inserted through the computation. Some continuous-time Diffusion Models [Campbell et al., 2023] can increase the entries in the data point during the reverse process, but can not remove them, making them unsuitable for property optimization. At the same time, they are better suited for continuous data with positional dependencies, rather than graphs.

# 3 Graph Insert-Delete Discrete Diffusion

We now introduce our main contribution. The backbone of GRIDDD is a reformulation of the standard discrete denoising diffusion for graphs to support the use of monotonic node insertions and deletions, which we describe in this section.

**Preliminaries.** Our end goal is to obtain a denoised graph $G^0 = (X^0, E^0)$ with $n^0$ nodes from a latent noisy graph with $G^T = (X^T, E^T)$ with $n^T$ nodes. To change size from $n^T$ to $n^0$, we *monotonically* delete (if $n^T > n^0$) or insert (if $n^T < n^0$) $|\Delta^T|$ nodes, where $\Delta^T = n^T - n^0$, as the reverse process advances. This gives us full control over the graph size at any given moment. The timesteps when insertions and deletions are performed are sampled from the normalization of the absolute value of the logistic distribution's probability density function $\zeta'(t)$:

$$\zeta'(t) = \left| \frac{e^{\frac{t-D}{w}}}{w(e^{\frac{t-D}{w}} + 1)^2} \right|. \tag{1}$$

The function is scaled in a way such that $\sum_{t=0}^{T} \zeta'(t) = 1$, with $\zeta'(0) = \zeta'(T) = 0$. The parameter $D$ indicates the timestep in which the function is maximized, while $w$ is the scale parameter (from a visual standpoint, it controls the steepness of the curve). During training, the final size $n^T$ that a training sample with size $n^0$ will assume is sampled from a discrete distribution on the number of nodes $h_{n^0}(n)$. In our work, we defined it as follows, and then normalized it to sum to one:

$$h_{n^0}(n) = p_{\max} + \frac{p_{\min} - p_{\max}}{n_{\max}} |n - n^0|, \tag{2}$$

where $n_{\max}$ is the maximum size that a graph may assume (which can be set to be higher than the size of the largest graph in the dataset), while $p_{\min}$ and $p_{\max}$ are hyperparameters. Intuitively, the highest probability $p_{\max}$ occurs when $n = n^0$, and it linearly decreases to $p_{\min}$ with a rate equal to $\frac{p_{\min} - p_{\max}}{n_{\max}}$ as the absolute difference between $n^0$ and $n$ increases.

## 3.1 Forward process

The discussion above implies that we have three different forward processes, depending on whether $\Delta^T > 0$, $\Delta^T < 0$, or $\Delta^T = 0$. We start by describing the latter, which is the simplest and serves as a basis for the deletion and insertion cases. Building on previous works [Vignac et al., 2023a, Ninniri et al., 2024], when $\Delta^T = 0$, the forward process is implemented as:

$$q(G^t \mid G^{t-1}) = (X^{t-1} Q_X^t, E^{t-1} Q_E^t). \tag{3}$$

Focusing on the nodes $X$ (edges are defined analogously), the associated transition matrix is:

$$Q_X^t = \alpha^t A_X + (1 - \alpha^t) B_X, \tag{4}$$

where $A_X = I$, $B_X = 1_a m_X^\intercal$, with $m_X$ being the vector of the marginal distribution of the atom types computed on the training set, and $\{\alpha_t\}_{i=1...T}$ being a set of *noise schedulers*, chosen such that $\alpha_1 = 1$, $\alpha_T = 0$, and the intermediate elements gradually decrease from one to zero. Multi-step transitions from an arbitrary timestep $s$ can be computed as a matrix $\overline{Q}_X^{t|s}$ defined as:

$$\overline{Q}_X^{t|s} = \prod_{i=s+1}^{t} Q_X^i = \overline{\alpha}^{t|s} A_X + (1 - \overline{\alpha}^{t|s}) B_X, \tag{5}$$

where $\overline{\alpha}^{t|s} = \frac{\overline{\alpha}^{t|0}}{\overline{\alpha}^{s|0}}$, with $\overline{\alpha}^{t|0} = \prod_{i=1}^{t} \alpha^i = \cos\left(\frac{\pi}{2} \frac{(\frac{t}{T} + s)^\nu}{1+s}\right)^2$ (notice the hyperparameter $\nu$).

**Monotonic deletions ($\Delta^T < 0$).** To introduce deletions, we build on Johnson et al. [2021] and treat them as an additional atom type DEL. This is similar to absorbing diffusion [Kong et al., 2023], in which the forward process consists of progressively switching all atom types to the deletion type; our strategy is more general, as we allow for nodes and edges to change category during the forward and reverse processes. During the reverse process, nodes need to be reinserted at step $t$ whenever they were deleted during the forward process at the same timestep (that is, if they transitioned to type DEL at forward step $t$). Consequently, such nodes are expected to revert to a

$$
\boldsymbol{A}^* =
\left[
\begin{array}{ccccc:c:c}
 & & & & & 0 & 0 \\
 & \boldsymbol{A} & & & & \vdots & \vdots \\
 & & & & & 0 & 0 \\
\hdashline
0 & \cdots & 0 & & 0 & 1 & 0 \\
\hdashline
0 & \cdots & 0 & & 0 & 1 & 0 \\
\end{array}
\right]
\begin{array}{l} \\ \\ \\ \texttt{D} \\ \texttt{D}^* \end{array}
\qquad (1)
$$

$$
\boldsymbol{B}^* =
\left[
\begin{array}{ccccc:c:c}
 & & & & & 0 & 0 \\
 & \boldsymbol{B} & & & & \vdots & \vdots \\
 & & & & & 0 & 0 \\
\hdashline
0 & \cdots & 0 & & 0 & 1 & 0 \\
\hdashline
0 & \cdots & 0 & & 0 & 1 & 0 \\
\end{array}
\right]
\begin{array}{l} \\ \\ \\ \texttt{D} \\ \texttt{D}^* \end{array}
\qquad (2)
$$

$$
\boldsymbol{C}^* =
\left[
\begin{array}{ccccc:c:c}
0 & \cdots & 0 & & 0 & 0 & 1 \\
\vdots & \ddots & \vdots & & \vdots & \vdots & \vdots \\
0 & \cdots & 0 & & 0 & 0 & 1 \\
\hdashline
0 & \cdots & 0 & & 0 & 1 & 0 \\
\hdashline
0 & \cdots & 0 & & 0 & 1 & 0 \\
\end{array}
\right]
\begin{array}{l} \\ \\ \\ \texttt{D} \\ \texttt{D}^* \end{array}
\qquad (3)
$$

$$
\boldsymbol{D}^* =
\left[
\begin{array}{ccccc:c:c}
0 & \cdots & 0 & & 0 & 1 & 0 \\
\vdots & \ddots & \vdots & & \vdots & \vdots & \vdots \\
0 & \cdots & 0 & & 0 & 1 & 0 \\
\hdashline
0 & \cdots & 0 & & 0 & 1 & 0 \\
\hdashline
0 & \cdots & 0 & & 0 & 1 & 0 \\
\end{array}
\right]
\begin{array}{l} \\ \\ \\ \texttt{D} \\ \texttt{D}^* \end{array}
\qquad (4)
$$

Figure 2: The matrices employed in the computation of $\boldsymbol{Q}^{*t}$ and $\overline{\boldsymbol{Q}}^{*t|s}$. For spacing reasons, the states DEL and DEL$^*$ have been shortened as D and D$^*$.

proper (non-DEL) type at the next step $t-1$. However, since the DEL state is absorbing (meaning $p(\boldsymbol{x}^t \neq \texttt{DEL} \mid \boldsymbol{x}^{t-1} = \texttt{DEL}) = 0$), such transition is forbidden: once a node is deleted, it must remain deleted up to the end of the forward process, as doing otherwise would violate monotonicity. To enable node reinsertion, we introduce an auxiliary transient type DEL$^*$, designed such that $p(\boldsymbol{x}^t = \texttt{DEL}|\boldsymbol{x}^{t-1} = \texttt{DEL}^*) = 1$, i.e., the transition from DEL$^*$ to DEL during the forward process is deterministic, and $p(\boldsymbol{x}^{t-1} \in \{\texttt{DEL}, \texttt{DEL}^*\}|\boldsymbol{x}^t = \texttt{DEL}^*) = 0$. This ensures that if a node is in state DEL$^*$ during the reverse process, it can only switch to a proper (non-DEL) type. This mechanism guarantees that nodes deleted at step $t$ can be reinserted at step $t-1$, thereby preventing the absorbing DEL state from blocking node recovery. To ensure that the final graph size is consistent with the one fixed at the start of the forward process, preventing that too many nodes switch into the deletion state, we employ a hybrid scheme, in which $n_0 - |\Delta^T|$ nodes undergo the standard forward process (described by Eq. 4), while $|\Delta^T|$ nodes use a different transition matrix[1] defined as:

$$
\boldsymbol{Q}^{*t} = \zeta(t)\left(\alpha^t \boldsymbol{A}^* + (1-\alpha^t)\boldsymbol{B}^*\right) + (1-\zeta(t))\,\boldsymbol{C}^*.
\qquad (6)
$$

Above, $\boldsymbol{A}^*$ and $\boldsymbol{B}^*$ are $\boldsymbol{A}$ and $\boldsymbol{B}$ augmented to account for the deletion types, $\boldsymbol{C}^*$ describes the transition from a normal type to a deletion (see Figure 2), and $\zeta(t)$ is the integral of $\zeta'(t)$ described in Equation 1. Since $\zeta'(t)$ is a probability distribution, $\zeta(t)$ can be seen as its cumulative distribution function; therefore, at time $T$, $\boldsymbol{Q}^{*T} = \boldsymbol{C}^*$. If $\zeta'$ is chosen such that $\zeta(T-1) = 0$, we enforce exactly $|\Delta^T|$ nodes with DEL type at timestep $T$. We refer the reader to Appendix A.1 for more details on the function $\zeta'$ used in this study. Computing the cumulative transition matrix $\overline{\boldsymbol{Q}}^{*t|s} = \prod_{i=s+1}^{t} \boldsymbol{Q}^{i*}$ requires an additional matrix $\boldsymbol{D}^*$ (also displayed in Figure 2):

$$
\overline{\boldsymbol{Q}}^{*t|s} = \overline{\zeta}^{t|s}\left(\overline{\alpha}^{t|s}\boldsymbol{A}^* + \left(1 - \overline{\alpha}^{t|s}\right)\boldsymbol{B}^*\right) + \overline{\zeta}^{t-1|s}\left(1 - \zeta(t)\right)\boldsymbol{C}^* + \left(1 - \overline{\zeta}^{t-1|s}\right)\boldsymbol{D}^*,
\qquad (7)
$$

where $\overline{\zeta}^{t|s} = \prod_{i=s+1}^{t} \zeta(i)$. Intuitively, $\boldsymbol{D}^*$ represents a state where all nodes are immediately switched to state DEL. If a node is set to category DEL or DEL$^*$, so are the edges associated with it.

**Monotonic insertions ($\Delta^T > 0$).** Node insertions are defined implicitly (i.e., without a specific INS type) since by design, a node is effectively inserted in the graph, or "activated", only at timestep $s+1$ if the model sampled $s$ as insertion time. The critical operation after inserting nodes is to establish which category they belong to. Our solution is to sample the inserted node's state from the marginal distribution $\boldsymbol{m}_X$ of the graph itself, which can be precomputed with no significant computational overhead (for more information, refer to Appendix A.2). Once a node is inserted, so are the edges connecting it to the rest of the graph. Their initial state is computed analogously using $\boldsymbol{m}_E$.

---

[1]From here on, we drop the subscript $\boldsymbol{X}$ for brevity. Equations 6–9 are defined analogously for $\boldsymbol{E}$.

## 3.2 Reverse process

The reverse process starts from a randomly initialized graph and gradually removes noise. However, since our main objective is to perform conditional generation, we also input the conditioning vector $\boldsymbol{y}$ using classifier-free guidance [Ho and Salimans, 2022]. We factorize the reverse process as the product of the individual reverse probabilities of the nodes and edges (assuming mutual independence, which we discuss in Appendix B.4):

$$p_\theta(\boldsymbol{G}^{t-1}|\boldsymbol{G}^t, \boldsymbol{y}) = \prod_{i \in 0,\ldots,n^t-1} p_\theta(\boldsymbol{x}_i^{t-1}|\boldsymbol{x}_i^t, \boldsymbol{y}) \prod_{i,j \in 0,\ldots,n^t-1} p_\theta(\boldsymbol{e}_{ij}^{t-1}|\boldsymbol{e}_{ij}^t, \boldsymbol{y}). \qquad (8)$$

In previous works [Ninniri et al., 2024], the posterior for the nodes was implemented with the aid of a neural network $p_\theta(\boldsymbol{x}^0 = \boldsymbol{x}|\boldsymbol{x}^t, \boldsymbol{y})$ tasked to predict the true atom and bond types given their noisy categories and the guide. However, this no longer works in our setting, as we cannot predict nodes that did not exist when $t = 0$ but were inserted subsequently during the forward process. To address this issue, we let the main model predict, alongside $\boldsymbol{G}^0$, the activation time $\hat{s}$ of each node and introduce the following generalized posterior:

$$p_\theta(\boldsymbol{x}^{t-1}|\boldsymbol{x}^t, \boldsymbol{y}) = \sum_{\boldsymbol{x} \in \chi} \frac{q(\boldsymbol{x}^t|\boldsymbol{x}^{t-1})q(\boldsymbol{x}^{t-1}|\boldsymbol{x}^{\hat{s}} = \boldsymbol{x})}{q(\boldsymbol{x}^t|\boldsymbol{x}^{\hat{s}} = \boldsymbol{x})} \, p_\theta(\boldsymbol{x}^{\hat{s}} = \boldsymbol{x}|\boldsymbol{x}^t, \boldsymbol{y}). \qquad (9)$$

The activation time of the nodes belonging to the original graph is assumed to be zero. Notice that when $q(\boldsymbol{x}^t|\boldsymbol{x}^0 = \boldsymbol{x}) = 0$, then $p_\theta(\boldsymbol{x}^{t-1}|\boldsymbol{x}^t, \boldsymbol{y})$ is set to zero as well. The edge posterior is computed analogously. Below, we describe how to extend the base setting to support insertions and deletions.

**Deletions.** During the reverse process, nodes deleted during the forward process need to be re-inserted. We do so with an auxiliary neural network $g_\phi(\boldsymbol{G}^t)$ that predicts, given the latent noisy graph $\boldsymbol{G}^t$, the number of DEL*s to add before passing the model to the main neural network that predicts the original graph $\boldsymbol{G}^0$ given $\boldsymbol{G}^t$. The auxiliary network is trained alongside the main model by masking out the nodes set to DEL* in $\boldsymbol{G}^t$ after computing $p_\theta(\boldsymbol{G}^0|\boldsymbol{G}^t, \boldsymbol{y})$, which reduces the training time significantly. Notice that $g_\phi$ is trained only when $\zeta'(t) > 0$, since it is not possible to insert nodes otherwise.

**Insertions.** Conversely to the deletion case, nodes inserted during the forward process need to be deleted. To do this, we let the main model predict, alongside $\boldsymbol{G}^0$, the activation time $\hat{s}$ of each node. Then, at timestep $t$, we remove from the graph the nodes predicted to have been inserted at step $t$. The activation time of the nodes belonging to the original graph is assumed to be zero.

## 3.3 Training and sampling

**Training.** During the learning phase, our goal is to train the model by corrupting each dataset sample $\boldsymbol{G}^0 = (\boldsymbol{X}^0, \boldsymbol{E}^0)$ into $\boldsymbol{G}^{*t} = (\boldsymbol{X}^{*t}, \boldsymbol{E}^{*t})$, with $t \leq T$, sampled from a uniform distribution, using the forward process. Having been subject to insertions or deletions, $\boldsymbol{X}^{*t}$ might contain only a subset of the nodes in $\boldsymbol{X}^0$ (if we deleted nodes) or include a new set of nodes (the ones that have been inserted up to step $t$). We train the main model to predict the values that the nodes $\boldsymbol{X}^{*t}$ and edges $\boldsymbol{E}^{*t}$ had at their respective activation times $\boldsymbol{S}^*$ (which may be equal to zero if the element belongs to the original graph) and denote these prediction as $\widehat{\boldsymbol{X}}^{*0}$ and $\widehat{\boldsymbol{E}}^{*0}$. At the same time, we predict the activation times themselves, which we denote as $\widehat{\boldsymbol{S}}^*$. Finally, we train the auxiliary model $g_\phi(\boldsymbol{G}^{*t})$ to predict the correct amount $n_{\text{DEL}^*}$ of DEL*s that were present in $\boldsymbol{G}^{*t}$, which we denote as $\widehat{n}_{\text{DEL}^*}$. The final loss is a sum of Cross-Entropy (CE) terms of the targets $\boldsymbol{X}^{*t}, \boldsymbol{E}^{*t}, \boldsymbol{S}^*$, and $n_{\text{DEL}^*}$ against their respective predictions:

$$\lambda_X \text{CE}(\widehat{\boldsymbol{X}}^{*0}, \boldsymbol{X}^{*0}) + \lambda_E \text{CE}(\widehat{\boldsymbol{E}}^{*0}, \boldsymbol{E}^{*0}) + \lambda_S \text{CE}(\widehat{\boldsymbol{S}}^*, \boldsymbol{S}^*) + \lambda_{\text{DEL}} \text{CE}(\widehat{n}_{\text{DEL}^*}, n_{\text{DEL}^*}), \qquad (10)$$

where $\lambda_X, \lambda_E, \lambda_E$, and $\lambda_{\text{DEL}}$ are hyperparameters that control the weight of each term. If we are training a model to condition on a specific set of properties $\boldsymbol{y}$, it is also fed at training time an auxiliary term $\boldsymbol{c}$ which, with a probability equal to $1 - \rho$, $\rho$ being an hyperparameter, will be equal to $\boldsymbol{y}$, while in the remaining cases it will assume the value of a parametrized placeholder $\overline{\boldsymbol{y}}$, which is randomly initialized. This technique is called *conditional dropout* and is shown to produce better results in conditional generation [Ho and Salimans, 2022]. Algorithm 1 in Appendix A.3 describes the new training process.

**Sampling.** During sampling, we start from a random latent $G^T$ sampled from the marginal distributions on the nodes and edges $m_X$ and $m_E$. The size of a sample is also chosen from the marginal distribution of the nodes in the training set. At each step, we first insert as many DEL* nodes as predicted by the auxiliary neural network. Then, we feed the current latent $G^t$ to the main model to predict $G^{*0}$. That is, the values that each node and edge had at their respective activation timesteps, as well as the activation timesteps themselves. Afterwards, we delete the nodes that are predicted to have been inserted at timestep $t$. Finally, we use these information to compute Equation 9, and sample from it the node and edge values at step $t-1$ to obtain $G^{t-1}$. We then repeat the operations using it as the new $G^t$, and we keep doing so for a total of $T$ steps until we obtain the final graph $G^0$. In the case in which we are performing conditional generation on a property $y$, the term $p_\theta(x^{\hat{s}}|x^t)$ is re-elaborated as follows:

$$p_\theta(x^{\hat{s}} = x|x^t, y) = p_\theta(x^{\hat{s}} = x|x^t, \overline{y}) + \lambda(p_\theta(x^{\hat{s}} = x|x^t, y) - p_\theta(x^{\hat{s}} = x|x^t, \overline{y})), \quad (11)$$

where $\lambda$ is a hyperparameter controlling the influence of the conditioning vector $y$. Algorithm 2 in Appendix A.3 describes the sampling process in detail.

## 4 Experiments

Here, we detail the experiments to evaluate GRIDDD on different property targeting and optimization tasks. Our code is based on MiDi [Vignac et al., 2023b] which, in turn, is based on DiGress. In all our experiments, we have $T = 500$ diffusion timesteps. The function $\zeta'(t)$ is parameterized with $w = 0.05$ and $D = \frac{T}{2} = 250$. Similarly to MiDi, we use $\nu = 1$ for the node matrices' noise schedulers and $\nu = 1.5$ for the edge matrices. $\lambda_X$ and $\lambda_E$ are set, respectively, to 1 and 2. We set the hyperparameters $p_{\min}$ and $p_{\max}$ in Equation 2 respectively as 0.2 and 1.

**Datasets.** Following previous works [Ninniri et al., 2024, Vignac et al., 2023a], we used QM9 [Ramakrishnan et al., 2014], a dataset of 133k molecules made by up to 9 non-hydrogen atoms, and ZINC-250k, a collection of 250k drug-like molecules selected from the ZINC dataset [Irwin and Shoichet, 2005]. On ZINC-250k, the molecules were first preprocessed by removing stereochemistry information and infrequent non-neutrally-charged atoms, leaving only N+ and O-. These atoms have been treated as standalone atom types. Dataset splits are described in Appendix B.1.

### 4.1 Property Targeting

We adhere to the setup of Ninniri et al. [2024], where they extract 100 property vectors from the test set and generate, for each one of them, 10 molecules. Then, they compute the Mean Absolute Error (MAE) between the target properties $y$ and the estimated properties of the generated molecules $\widehat{y}$:

$$\text{MAE}(y, \widehat{y}) = \frac{1}{1000} \sum_{i=1}^{100} \sum_{j=1}^{10} |y_i - \widehat{y}_{i,j}|. \quad (12)$$

On QM9, we targeted the Dipole Moment $\mu$ and the Highest Occupied Molecular Orbital (HOMO) properties, while on ZINC-250k we targeted the Log-Partition coefficient (LogP), the Quantitative Estimation of Drug-likeness (QED), and the molecular weight (MW). Since MW strongly correlates with the number of nodes of the molecular graph, it provides a natural and intuitive testbed to assess the ability to flexibly adapt the graph size, which is precisely what GRIDDD was designed for. Our baseline for comparison is discussed in Appendix B.3.

### 4.2 Property Optimization

We use the test suite by Jin et al. [2020], which consists of three experiments on the ZINC-250k dataset. The objective is to generate molecules that improve on a certain property while meeting a similarity constraint. To optimize a molecule, we corrupt it for 100 steps before starting the denoising process, setting the property vector $y$ as the target value that the optimized molecule should have. In the **LogP** experiment, we optimize the 800 molecules with the lowest LogP among those in the test set. For each of them, we sample 20 candidates starting from different latents and only keep the one with the largest improvement among the ones with a fingerprint Tanimoto similarity with the starting molecule equal to or above a threshold $\delta$. We report the average of these improvements

across the 800 molecules, with $\delta \in \{0.4, 0.6\}$. In the **QED** experiment, we optimize 800 molecules from the test set with QED in the range $[0.7, 0.8]$. For each molecule, we sample 20 candidates from different latents. We consider the optimization successful if at least one among the candidates has a QED in the range $[0.9, 1.0]$ and a fingerprint Tanimoto similarity with the starting molecule $\geq 0.4$. We report the success rate across the 800 molecules. In the **DRD2** experiment, we optimize for the biological activity against the dopamine type 2 receptor, selecting 800 molecules from the test set with DRD2 activity score $\leq 0.05$. For each molecule, we sample 20 candidates from different latents. We consider the optimization successful if at least one among the candidates has a DRD2 score $\geq 0.5$ and a fingerprint Tanimoto similarity with the starting molecule $\geq 0.4$. We report the success rate across the 800 molecules. Our baseline for comparison is discussed in Appendix B.3.

### 4.3 Out-of-distribution sampling

To assess whether GRIDDD is able to generate valid molecules outside the training distribution, we train an instance of DiGress and one of GRIDDD to generate samples unconditionally on QM9. However, during training, we force GRIDDD to *insert* up to 14 nodes in the molecular graph, despite the fact that QM9 features molecules with up to 9 atoms. Then, during generation, we force both models to generate molecules with up to 15 atoms (that is, above GRIDDD's training capacity).

**Experimental setup.** The denoising network shares a similar architecture to FreeGress, and is described in Appendix B.2. For QM9, we use the same hyperparameters employed by the authors. The auxiliary neural network used to predict the number of DEL*s also uses the exact same setup, with the exception that it only uses one layer. For ZINC-250k, we also use the same hyperparameters as FreeGress except for the number of layers, which we set to 10 instead of the original 12 to use approximately the same number of parameters. The size of the linear layer used to predict the activation time is 256. In QM9, we set the guidance scale to $\lambda = 3$, while in ZINC-250k we set $\lambda = 2$. These values have been chosen since they were the ones offering the best results for FreeGress. All experiments have been performed on an nVidia A100 GPU with 80 GBs of VRAM (two on ZINC-250k).

## 5 Results

Here, we detail the experimental results obtained by GRIDDD. A visual collection of samples obtained from the model is available in Appendix D.3

### 5.1 Property targeting

**QM9.** Results are displayed in Table 1. As can be seen, GRIDDD stably achieves state-of-the-art performance. When conditioning on $\mu$, it improves MAE by 14% at the expense of a slightly lower validity. Conversely, when conditioning on HOMO, the model achieves a better validity at the expense of a slightly higher MAE. For ablation purposes, we also include an unconditional model in the comparison, which is unable to optimize the properties to a satisfactory degree. A discussion of the failure cases in QM9 is presented in Appendix D.4.

Table 1: Results of property targeting on QM9.

| Method | $\mu$ | | HOMO | |
|---|---|---|---|---|
| | **MAE** $\downarrow$ | **Val.** $\uparrow$ | **MAE** $\downarrow$ | **Val.** $\uparrow$ |
| Unconditional | $1.68 \pm 0.15$ | $91.5\%$ | $0.95 \pm 0.10$ | $91.5\%$ |
| DiGress | $0.80 \pm 0.07$ | $82.5\%$ | $0.61 \pm 0.07$ | $91.2\%$ |
| FreeGress | $0.74 \pm 0.08$ | $\mathbf{83.7}\%$ | $\mathbf{0.32 \pm 0.04}$ | $90.1\%$ |
| GRIDDD | $\mathbf{0.66 \pm 0.07}$ | $79.0\%$ | $0.37 \pm 0.07$ | $\mathbf{94}\%$ |

**ZINC-250k.** Results are shown in Table 2. On ZINC-250k, GRIDDD once again achieves comparable or better MAE, while improving validity rates in two cases out of three. Of particular interest is the MW experiment, as this property strongly correlates with the graph size. Note that FreeGress'

results have been obtained using an auxiliary neural network to predict the optimal number of atoms the molecule should have given the target MW, information which is used to set the graph's size before starting the denoising process. GRIDDD is capable of halving FreeGress' MAE without any prior knowledge of the graph size distribution. In summary, GRIDDD shows an improved capacity in generating valid molecules without sacrificing the MAE. What is truly relevant in these results is the fact that GRIDDD is capable of obtaining them despite being trained on a more difficult task, showing that our solution has a smoother tradeoff between validity and accuracy.

Table 2: Results of property targeting on the ZINC-250k dataset.

| Method | LogP | | QED | | MW | |
|---|---|---|---|---|---|---|
| | MAE ↓ | Val. ↑ | MAE ↓ | Val. ↑ | MAE ↓ | Val. ↑ |
| Unconditional | $1.52 \pm 0.12$ | 86.1 % | $0.15 \pm 0.01$ | 86.1 % | $74.16 \pm 6.71$ | 86.1 % |
| DiGress | $0.74 \pm 0.08$ | 74.6 % | $0.15 \pm 0.01$ | 85.1 % | $20.92 \pm 2.90$ | 40.4 % |
| FreeGress | $\mathbf{0.17 \pm 0.01}$ | 84.9 % | $0.04 \pm 0.01$ | 84.9 % | $8.96 \pm 1.93$ | 79.7 % |
| GRIDDD | $0.19 \pm 0.02$ | **87.9** % | $\mathbf{0.04 \pm 0.00}$ | **87.2** % | $\mathbf{4.89 \pm 0.57}$ | **84.2**% |

## 5.2 Out-of-distribution sampling

Results are displayed in Figure 3. While both models perform similarly with up to 12 atoms, the validity rates of DiGress steeply decrease on larger graph sizes. Notably, GRIDDD still generates 35% of valid molecules with 15 nodes, despite being trained on latents with up to 14 nodes. This shows how our model is fairly usable in out-of-distribution sampling, as it is able to learn graph sizes that do not appear in the training dataset.

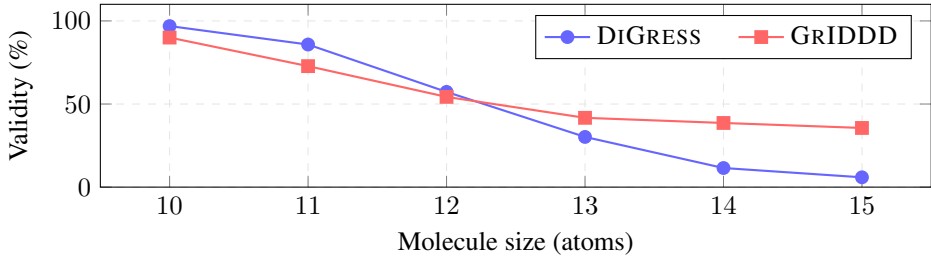

Figure 3: Validity results of out-of-distribution sampling on QM9.

## 5.3 Property optimization

Results are summarized in Table 3. Regarding the LogP property optimization, GRIDDD outperforms all competitors in both settings in terms of improvement. In particular, when the similarity threshold is 0.6, it achieves an average improvement almost twice as high as the second best. In the other two tasks, GRIDDD scores remarkably better, especially in the QED task where it improves 5 times the success rates of GCPN. An additional comparison with the method by Ketata et al. [2025] (under a slightly different experimental setup) is provided in Appendix D.1, showing that GRIDDD performs comparably or better than methods that use 3D information to optimize molecules.

## 5.4 Computational considerations

The insertions and deletions performed by GRIDDD inevitably introduce computational overhead during training, leading to slightly increased training durations compared to methods that do not employ these operations. In particular, our analysis indicates that it is 30% slower than FreeGress (see Appendix C.1 for an extended discussion). However, its flexibility in adapting the number of denoising steps to the task makes it generally faster to sample from (as detailed in Appendix C.2).

**Ablation.** To test the effective influence of insert and delete operations over the optimization process, we performed the same experiments with GRIDDD but disabling insertions and deletions.

Table 3: Benchmark on graph property optimization.

| Method | LogP (sim $\geq$ 0.4) | | LogP (sim $\geq$ 0.6) | | QED (sim $\geq$ 0.4) | | DRD2 (sim $\geq$ 0.4) | |
|---|---|---|---|---|---|---|---|---|
| | Improv. ↑ | Div.↑ | Improv. ↑ | Div.↑ | Succ.↑ | Div.↑ | Succ.↑ | Div.↑ |
| JT-VAE | $1.03 \pm 1.39$ | - | $0.28 \pm 0.79$ | - | 8.8 % | - | 3.4 % | - |
| CG-VAE | $0.61 \pm 1.09$ | - | $0.25 \pm 0.74$ | - | 4.8 % | - | - | - |
| GCPN | $2.49 \pm 1.30$ | - | $0.79 \pm 0.63$ | - | 9.4 % | 0.216 | 4.4 % | **0.152** |
| GRIDDD | $\mathbf{2.70 \pm 0.94}$ | 0.482 | $\mathbf{1.33 \pm 0.61}$ | 0.280 | **45.1** % | **0.283** | **5.0** % | 0.121 |

The results show that while the success rate remains relatively unchanged in the LogP and DRD2 experiments, it significantly drops to 33.8% when optimizing QED, likely because the QED score is a function of the molecular weight (and thus, it correlates with the number of atoms). We conclude that, similarly to targeting MW, controlling the graph size substantially increases success rates.

A naive form of insertion and deletion operations can be achieved implicitly on any graph DDPM by working with a large graph size (even beyond the largest in the training data), treating excess nodes/edges as padding nodes marked by a special PAD class which allows their removal at the end of the denoising process. Clearly, this method is slower to train and sample from, as it always works with maximally sized graphs. To check that this approach is also less accurate, we compared two DiGress variants that implement node and edge padding against GRIDDD: the first only assigns the PAD category to padding nodes, while the second also assigns it to their associated edges. All models were trained on the QM9 dataset for 250 epochs, after which we sampled 100 molecules each without conditioning. The results are displayed in Table 4. As it is possible to see, while the baseline generates 3% more valid molecules than GRIDDD, they severely underperform in every other metric.

Table 4: Comparison of GrIDDD with DiGress-based padding variants that perform implicit insertions and deletions. Val: validity, Avg NC: average number of connected components, Max NC: maximum number of connected components, NSC: number of graphs sampled with a single connected component, XCE (ECE): validation cross-entropy over nodes (edges).

| Model | Val ↑ | Avg NC ↓ | Max NC ↓ | NSC ↑ | XCE ↓ | ECE ↓ |
|---|---|---|---|---|---|---|
| GrIDDD | 0.97 | **1** | **1** | **100** | **0.44** | **0.32** |
| Nodes PAD | 1 | 1.56 | 8 | 69 | 0.47 | 0.37 |
| All PAD | 1 | 1.2 | 4 | 84 | 0.48 | 0.38 |

In Appendix D.2, we conduct a sensitivity analysis on some critical hyperparameters of GRIDDD, showing that results are robust to their choices.

# 6 Conclusions

We introduced GRIDDD, a discrete DDPM which supports the insertion and deletion of atoms during the denoising process. This addresses one of the major limitations of graph DDPMs, which cannot change the graph size throughout the generative process. In future studies, we aim to study a simplified approach that does not need an auxiliary network to predict when to insert a node during the denoising process, and whether the distribution of the inserted nodes can be learned rather than employing predefined heuristics. Lastly, since the methodology we propose is quite general, we foresee its applications to domains such as, e.g., vector floor-plan generation [Shabani et al., 2023].

**Limitations.** During generation, we observed that the two neural networks composing GRIDDD may occasionally conflict, simultaneously performing an insertion and a deletion at the same timestep, which is theoretically illegal in our framework. GRIDDD also tends to produce more split molecules compared to FreeGress and DiGress. This is likely because nodes inserted late in the denoising process are challenging to connect before denoising concludes. Incorporating additional input features, such as the current number of connected components, could potentially mitigate this issue.

## Acknowledgments

This work has been partially supported by EU-EIC EMERGE (Grant No. 101070918) and PNRR, PE00000013, "FAIR - Future Artificial Intelligence Research", Spoke 1, funded by European Commission under NextGeneration EU programme (CUP: B53D2302625000).

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

## A  Additional details on the methodology

### A.1  Delete scheduler and insert/delete timestep distributions

We show a possible realization of the delete scheduler $\zeta(t)$ and the insert/delete timestep distribution $\zeta'(t)$ in Figure 4.

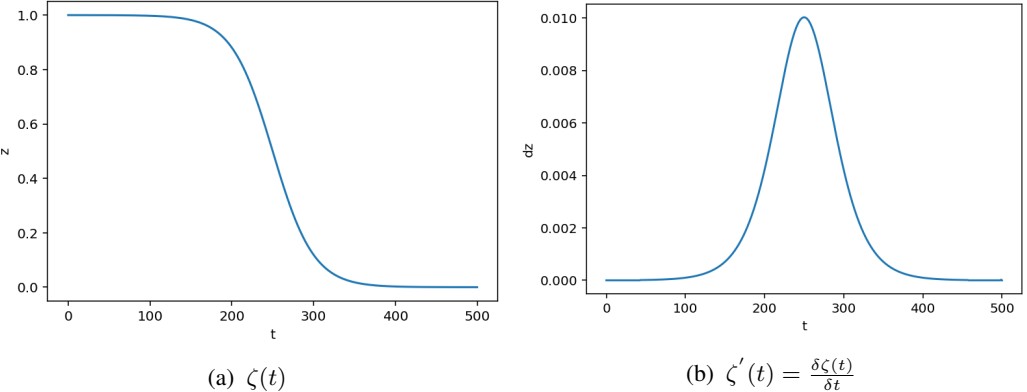

(a) $\zeta(t)$    (b) $\zeta'(t) = \frac{\delta\zeta(t)}{\delta t}$

Figure 4: The functions $\zeta(t)$ and its derivative $\zeta'(t)$ for $w = 0.05$ and $T = 500$. $1 - \zeta(t)$ is effectively the "delete scheduler", and represents the probability that a node selected for deletion has already been deleted at timestep $t$. $\zeta'(t)$ represents instead the probability that a node chosen for deletion will switch to category $\text{DEL}^*$ exactly at timestep $t$.

### A.2  Distribution of the labels of the inserted nodes and edges

During the forward process, GRIDDD assigns labels to the newly inserted nodes and edges using a marginal distribution of these labels computed on the individual training sample. Such distributions can be computed just once before starting the training process and can also be saved for multiple training sessions. We have also tried different distributions during our preliminary studies, such as always inserting the least frequent node/edge class, as well as sampling them from a uniform probability, but the sample's marginal distribution gave us better results.

### A.3  Training and sampling algorithms

Here, we provide the pseudocode describing the procedures to train (Algorithm 1) and sample from (Algorithm 2) GRIDDD.

## B  Additional experimental details

### B.1  Dataset splits

**QM9.**  On QM9, the training set is made of the first 100000 samples in the dataset. The test set is made of $10\%$ of the overall data, and the remaining data is used to make the validation set.

**ZINC-250k.**  The training set uses the first $80\%$ of the data. The remainder is equally split between the validation set and the test set.

### B.2  Neural architecture

Both the main model and the auxiliary model employed to predict the number of DELs are based on FreeGress' architecture. The model is composed of a stack of Graph Transformer layers. It receives as input the node features matrix $\boldsymbol{X}$, edge features matrix $\boldsymbol{E}$, and a vector $\boldsymbol{u}$ encoding the timestep $t$ and optional extra features. These inputs are augmented with conditioning information $\boldsymbol{y}$ (denoted with the $+$ superscript in Figure 5a) and processed through the transformer stack.

---

**Algorithm 1** Pseudocode of the training algorithm

---

**Require:** $\mathcal{G} = \{\boldsymbol{G}_i\}_{i=1}^N$, dataset of $N$ graphs
**Require:** $T$, the number of denoising timesteps,
**Require:** $p_\theta$, the main model,
**Require:** $g_\phi$, the model tasked to predict the number of DEL$^*$s in $\boldsymbol{G}^t$,
**Require:** $\zeta'(t)$, a discrete probability distribution function defined on $T$,
**Require:** $h(n)$, which given a graph size $n$ returns a probability distribution $h_n$ on the number of
    nodes.
 1: **for each** graph $\boldsymbol{G}_i = (\boldsymbol{X}_i, \boldsymbol{E}_i) \in \mathcal{G}$: **do**
 2:    $n^0 = |\boldsymbol{X}_i|$
 3:    $n^T \sim h_{n^0}(n)$
 4:    $t \sim Uniform(1, T)$
 5:    $\boldsymbol{m}_{\boldsymbol{X}_i}, \boldsymbol{m}_{\boldsymbol{E}_i}$ = marginal distribution computed on the node and edges of $\boldsymbol{G}_i$.
 6:    $\boldsymbol{S}^* = \{0, \ldots, 0\}$ vector storing the insertion times of the nodes
 7:    $\boldsymbol{X}_i^{*t}, \boldsymbol{E}_i^{*t} \sim (\boldsymbol{X}_i^t)' \overline{Q}_{\boldsymbol{X}}^{t|0}, (\boldsymbol{E}_i^t)' \overline{Q}_{\boldsymbol{E}}^{t|0}$
 8:    $\Delta^T = n^T - n^0$
 9:    Sample $\boldsymbol{U} = \{u_j\}$ timesteps, with $j \in \{0 \ldots |\Delta^T| - 1\}$, where $u_j \sim \zeta'$
10:    Remove from $\boldsymbol{U}$ the elements greater than $t$
11:    **for each** $u_j \in \boldsymbol{U}$ **do**
12:        **if** $\Delta^T > 0$ **then**                                              ▷ Insert
13:            Sample a new node $\boldsymbol{x} \sim \boldsymbol{m}_{\boldsymbol{X}_i}$ with insertion time $u_j$, and add it to $\boldsymbol{X}_i$
14:            Sample $\boldsymbol{x}^t$ from the distribution $\boldsymbol{x}' \overline{\boldsymbol{Q}}_{\boldsymbol{X}}^{*t|u_j}$ and add it to $\boldsymbol{X}_i^{*t}$
15:            Sample from $\boldsymbol{m}_{\boldsymbol{E}_i}$, for each node already in the graph, an inward and
                outward edge with the newly added $\boldsymbol{x}_t$
16:            Sample the edges $\boldsymbol{e}$'s final state $\boldsymbol{e}^t$ from $\boldsymbol{e}' \overline{Q}_{\boldsymbol{E}}^{*t|u_j}$ and add them to $\boldsymbol{E}_i^{*t}$
17:            Append $u_j$ at the end of $\boldsymbol{S}^*$
18:        **else if** $\Delta^T < 0$ **then**                                         ▷ Delete
19:            Sample one random node $\boldsymbol{x}$ among the ones in $\boldsymbol{G}_i$ that are not set to state DEL
                or DEL$^*$
20:            **if** $u_j = t$ **then**
21:                $\boldsymbol{x} = $ DEL$^*$
22:                Set all incoming and outgoing edges of $\boldsymbol{x}$ as DEL$^*$
23:            **else**
24:                Remove $\boldsymbol{x}$ from $\boldsymbol{X}_i^{*t}$
25:                Remove all incoming and outgoing edges of $\boldsymbol{x}$ from $\boldsymbol{E}_i^{*t}$
26:                Remove the entry in $\boldsymbol{S}^*$ associated to the node
27:            **end if**
28:        **end if**
29:    **end for**
30:    $\boldsymbol{G}^{*t} = (\boldsymbol{X}^{*t}, \boldsymbol{E}^{*t})$
31:    $\{(\widehat{\boldsymbol{X}}^{*0}, \widehat{\boldsymbol{E}}^{*0}), \widehat{\boldsymbol{S}}^*\} = p_\theta(\boldsymbol{G}^{*t})$
32:    $loss_i = \lambda_X \text{CE}(\widehat{\boldsymbol{X}}^{*0}, \boldsymbol{X}^{*0}) + \lambda_E \text{CE}(\widehat{\boldsymbol{E}}^{*0}, \boldsymbol{E}^{*0}) + \lambda_S \text{CE}(\widehat{\boldsymbol{S}}^{*0}, \boldsymbol{S}^{*0})$
33:    $n_{\text{DEL}^*} = $ number of nodes set to DEL$^*$ in $\boldsymbol{G}^{*t}$
34:    Remove all nodes set to DEL$^*$ in $\boldsymbol{G}^{*t}$ as well as the associated edges
35:    $loss_i^* = CE(g_\phi(\boldsymbol{G}^{*t}), n_{\text{DEL}^*})$
36: **end for**

---

---

**Algorithm 2** Pseudocode of the denoising algorithm

---

**Require:** $T$, the number of denoising timesteps,
**Require:** $n^T$, the number of nodes in the noisy graph $\boldsymbol{G}^T$,
**Require:** $p_\theta$, the main model,
**Require:** $g_\phi$, the model tasked to predict the number of DEL\*s in $\boldsymbol{G}^t$,
**Require:** $\boldsymbol{m_X}$ and $\boldsymbol{m_E}$, the dataset's marginal distributions on the nodes and the edges;
1: Sample $\boldsymbol{G}^T = (\boldsymbol{X}^T, \boldsymbol{E}^T)$ with $n^T$ nodes from an the marginal distributions $\boldsymbol{m_X}$ and $\boldsymbol{m_E}$;
2: **for each** timestep $t \in [T, \ldots, 1]$: **do**
3:     Use $g_\phi(\boldsymbol{G}^t)$ to predict the number of DEL\* entries to insert in the graph, and insert them.
4:     Set all the edges connecting the newly inserted nodes to the rest of the graph as DEL\*.
5:     Use $p_\theta(\boldsymbol{G}^t)$ to predict $\boldsymbol{S}^t$, the insertion times for each node in the graph, and $\boldsymbol{G}^{*0}$.
6:     Remove the nodes $\boldsymbol{x}_i$ such that $\boldsymbol{S}_i = t$, as well as their edges.
7:     Use $\boldsymbol{S}^t$ and $\boldsymbol{G}^{*0}$ to sample, for each node $\boldsymbol{x}_i^t$ and edge $\boldsymbol{e}_{ij}^t$ in $\boldsymbol{G}^t$, their type at step $t-1$:
8:     $\boldsymbol{x}_i^{t-1} \sim p_\theta(\boldsymbol{x}_i^{t-1}|\boldsymbol{x}_i^t) = \sum_{x\in\chi} \frac{q(\boldsymbol{x}_i^t|\boldsymbol{x}_i^{t-1})q(\boldsymbol{x}_i^{t-1}|\boldsymbol{x}_i^{S_i}=x)}{q(\boldsymbol{x}_i^t|\boldsymbol{x}_i^{S_i}=x)} p_\theta(\boldsymbol{x}_i^{S_i} = x|\boldsymbol{x}^t).$
9:     $\boldsymbol{e}_{ij}^{t-1} \sim p_\theta(\boldsymbol{e}_{ij}^{t-1}|\boldsymbol{e}_{ij}^t) = \sum_{e\in\varepsilon} \frac{q(\boldsymbol{e}_{ij}^t|\boldsymbol{e}_{ij}^{t-1})q(\boldsymbol{e}_{ij}^{t-1}|\boldsymbol{e}_{ij}^{max(S_i,S_j)}=e)}{q(\boldsymbol{e}_{ij}^t|\boldsymbol{e}_{ij}^{max(S_i,S_j)}=e)} p_\theta(\boldsymbol{e}_{ij}^{max(S_i,S_j)} = e|\boldsymbol{e}_{ij}^t).$
10:    $\boldsymbol{G}^{t-1} = (\{\boldsymbol{x}_i^{t-1}\}, \{\boldsymbol{e}_{ij}^{t-1}\})$
11: **end for**

---

Each Graph Transformer layer (Figure 5b) follows the standard architecture: self-attention, dropout, residual connection, layer normalization, and a feedforward block. The self-attention mechanism (Figure 5c) applies scaled dot-product attention to the augmented node features $\boldsymbol{X}$. The resulting attention weights are modulated by the edge tensor $\boldsymbol{E}$ via a FiLM layer [Perez et al., 2018], normalized with softmax, and used to re-weight $\boldsymbol{X}$. The output is flattened and combined with $\boldsymbol{u}$ through another FiLM layer before linear projection to prediction logits.

Separately, $\boldsymbol{X}$ and $\boldsymbol{E}$ are passed through independent PNA layers [Corso et al., 2020], and their outputs are summed with a linearly projected $\boldsymbol{u}$ to form the final graph representation. In GRIDDD, the final output $\boldsymbol{u}^{'}$ is employed by our auxiliary model to predict the amount of DEL\* nodes that must be reintroduced at the current timestep. The final Graph Transformer Layer's output designed to predict the matrix $\boldsymbol{X}^{'}$ is passed to two Linear layers, instead of just one. The first is used, just like before, to predict the final node matrix. The second one is used instead to predict the activation timestep $s$ of the various nodes.

### B.3   Baselines

For the task of property targeting, discussed in Section 4.1, we compared our model against DiGress [Vignac et al., 2023a], a Graph Diffusion Model capable of performing conditional graph generation through a classifier-based guidance, and FreeGress [Ninniri et al., 2024], a Graph Diffusion Model which employs a classifier-free guidance system instead. For the task of property optimization, discussed in Section 4.2, we employ a baseline of three models. In particular, we employ the Junction-Tree Variational Autoencoder (JT-VAE, Jin et al. [2018a]), an Autoencoder that works on the molecular sub-structural level rather than the atomic level, the Graph Convolutional Policy Network (GCPN, You et al. [2018a]), which employs Reinforcement Learning to generate molecules atom-by-atom, and the Constrained Graph Variational Autoencoders (CG-VAE, Liu et al. [2018]), a graph-based Autoencoder whose peculiarity is that it applies chemical validity constraints during the generative process to increase its performance in molecular tasks.

### B.4   Considerations on the independence between nodes and edges in the forward process

When a node is switched to DEL\*, so are the edges associated with it. This effectively creates a dependence between the two, while standard Diffusion Models assume a mutual independence between them. This is similar to what is done with Masked Diffusion [Kong et al., 2023], where the edges associated with masked nodes are masked as well. In our case, what we effectively do is replacing the forward process $p(\boldsymbol{e}_{ij}^t|\boldsymbol{e}_{ij}^{t-1})$ with $p(\boldsymbol{e}_{ij}^t|\boldsymbol{e}_{ij}^{t-1}, \boldsymbol{x}_i^t, \boldsymbol{x}_j^t)$, and $p(\boldsymbol{e}_{ij}^{t-1}|\boldsymbol{e}_{ij}^t, \boldsymbol{e}_{ij}^0)$ with $p(\boldsymbol{e}_{ij}^{t-1}|\boldsymbol{e}_{ij}^t, \boldsymbol{e}_{ij}^0, \boldsymbol{x}_i^t, \boldsymbol{x}_j^t)$. However, it should be kept in mind how the latter equation is, in all practical

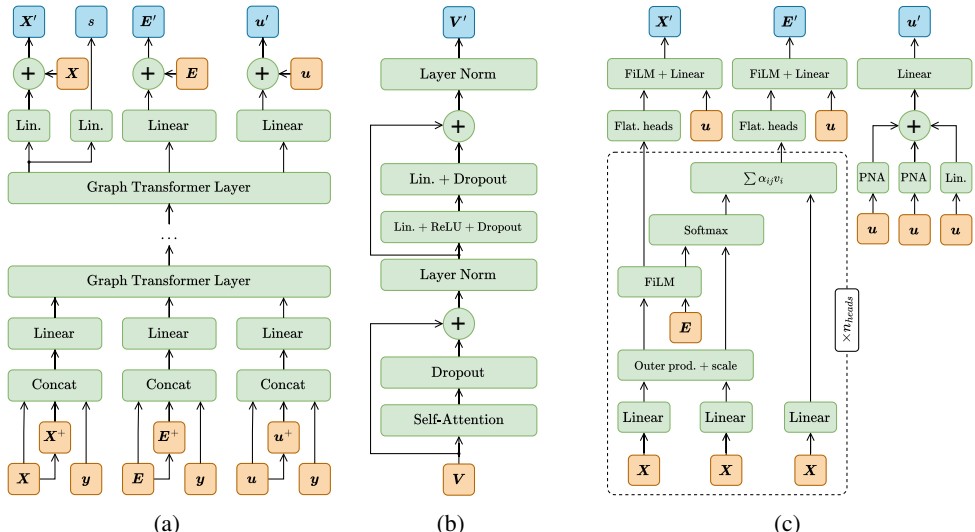

Figure 5: (a) High-level architecture of GRIDDD. (b) Detail of the graph transformer layer. $V$ represents the triple $(X, E, u)$. (c) The self attention-layer within the graph transformer layer. In the pictures, orange boxes indicate inputs, blue boxes indicate outputs, and green boxes indicate layers/operations.

scenarios, identical to $p(e_{ij}^{t-1}|e_{ij}^t, e_{ij}^0)$, as the only situations where $x_{i/j}^t$ could influence the value of $e_{ij}^{t-1}$ is when the former is labeled as DEL* and the latter to DEL, in which case $e_{ij}^{t-1}$ should switch to DEL* as well. However, this scenario is not possible in practice, as neither the nodes nor the edges can appear explicitly with label DEL during the reverse process. We leave to future research the study of independent edge processes that still allow for the deletion of the associated nodes.

## C  Additional computational considerations

### C.1  Runtime

On average, one training epoch of GRIDDD takes 309.1 seconds on two nVidia A100 GPUs. FreeGress, on the same problem (hence, without insert and delete operations), takes 243 seconds. This means that GRIDDD is approximately 27% slower than its counterpart. This is similar to the 33.3% increase we have found on QM9, where GRIDDD takes 18 seconds on average to complete a training epoch on a single A100, while FreeGress takes 13.5.

### C.2  Sampling speed

Interestingly enough, GRIDDD can be faster at sampling than standard Diffusion Models. When tasked with generating a variegated batch of graphs with different sizes, a standard Diffusion Model would first sample the graph sizes from the dataset's node distribution. This means that, in practice, the sizes of the node and edge matrices will have one or more dimensions equal to the highest graph size sampled. In turn, this means that every graph generated will require a processing power proportional to such size.

On the other hand, GRIDDD is capable of generating a graph of any shape, even when starting from a different size. In this scenario, one could start the sampling process with very small latents (even zero nodes). Since the tensors representing the sample batch would be very small, the computational power required to process these tensors will be small as well. This means that the early phases of the sampling process, before most nodes are inserted, will likely be faster than standard diffusion (although this is highly dependent on $\zeta(t)$).

To test this hypothesis, we have sampled 128 molecules on ZINC-250K, conditioning on the LogP value, with FreeGress and GRIDDD. For the latter, we have performed three experiments. In the first,

GRIDDD starts from latents with a size of two nodes. In the second, we have used an initial size of 24 (the most frequent graph size in the training set), and then we have sampled the initial graph size using the training set's node distribution, as is done in FreeGress. The results, summarized in Table 5, show how GRIDDD is capable of performing as well as FreeGress while halving the sampling time in the best case scenario, and even in the worst case scenario it requires as little as five percent more time to perform as well as FreeGress.

Table 5: Sampling speed comparison between GrIDDD and FreeGress when generating 128 samples on ZINC-250k conditioned on the LogP value, using different initial latent sizes.

| Model | Validity ↑ | MAE ↓ | Sampling time ↓ |
|---|---|---|---|
| FreeGress | 89.8% | 0.31 | 152.05s |
| GRIDDD (initial size=2) | **91.4%** | **0.23** | **68.09$s$** |
| GRIDDD (initial size=24) | 90.6% | 0.31 | 120.19s |
| GRIDDD (initial size $\sim p(n)$) | 88.2% | 0.26 | 159.85s |

# D  Additional results

## D.1  Comparison with EDM-SyCO

We compared to the 3D Diffusion Model by Ketata et al. [2025] called EDM-SyCO, since it performs property optimization similarly to GRIDDD. Notably, EDM-SyCO also inputs 3D coordinates to the diffusion process. To ensure a fair comparison, we slightly adapted our original setup to the less restrictive one used in the related paper. In particular, they attempt to optimize each molecule 100 times, rather than 20 as in our previous experiment. Then, among the ones within the similarity threshold, they select the 10 ones with the best improvement, duplicate each one of them ten times, and run the optimization process a second time. The process is repeated four times (for a total of 400 optimization rounds), after which the molecule with the best improvement within the similarity threshold is selected. All the successful molecules obtained in the process (that is, within the similarity threshold) are considered when computing the diversity score. In cases where the number of successful experiments is less than two, the diversity score is set to zero. Results of the comparison are reported in Table 6, showing competitive or superior performances by GRIDDD, even though it does not benefit from using 3D information as input. We have compared our results against EDM-SyCo on property targeting as well. Specifically, we have trained a model on ZINC-250K with 1000 denoising timesteps (as is done by our baseline) that is conditioned on the molecular weight. We have obtained a MAE of $2.13 \pm 0.19$ and a validity of $86.2\%$, while EDM-SyCo records a MAE of $3.86 \pm 0.08$ and a validity of $88\%$.

Table 6: Comparison with EDM-SyCo on property optimization.

| Method | LogP (sim $\geq$ 0.4) | | LogP (sim $\geq$ 0.6) | | QED (sim $\geq$ 0.4) | | DRD2 (sim $\geq$ 0.4) | |
|---|---|---|---|---|---|---|---|---|
| | Improv. ↑ | Div.↑ | Improv. ↑ | Div.↑ | Succ.↑ | Div.↑ | Succ.↑ | Div.↑ |
| EDM-SYCO | $3.11 \pm 1.27$ | **0.555** | $1.51 \pm 1.10$ | **0.360** | 46.4 % | 0.163 | **27.3 %** | **0.083** |
| GRIDDD | $\mathbf{3.27 \pm 0.91}$ | 0.511 | $\mathbf{1.59 \pm 0.54}$ | 0.359 | **64.1 %** | **0.269** | 19.7 % | 0.058 |

## D.2  Hyperparameter sensitivity analysis

We show in Table 7 an analysis of GRIDDD's sensitivity of the hyperparameters regulating the scheduler ($D$ and $w$) and the node counts distribution ($p_{\min}$ and $p_{\max}$). One interesting insight that can be gathered from the table is the fact that the number of split molecules generated significantly increases with smaller $D$s. From a practical perspective, small values for this hyperparameter imply that, during the denoising process, the model can insert nodes relatively late during the reverse process. We conjecture that split molecules are mostly caused by nodes inserted too late during such a process, as they cannot be re-attached in time to the main graph since the noise scheduler does not allow for too many changes in the graph's structure during the last phases of the process. We

have also noted that this phenomenon is sensibly reduced when GRIDDD is given in input, as extra features, different powers of the adjacency matrix.

Table 7: Study of GrIDDD's sensitivity to hyperparameter changes when sampling 100 molecules after training on the QM9 dataset. The hyperparameters tested are the delete scheduler's parameters $D, w$, and the node count distribution's parameters $p_{\min}$ and $p_{\max}$. Val: validity, Avg NC: average number of connected components, Max NC: maximum number of connected components, NSC: number of graphs sampled with a single connected component.

| $D$ | Val $\uparrow$ | Avg NC $\downarrow$ | Max NC $\downarrow$ | NSC $\uparrow$ | $w$ | Val $\uparrow$ | Avg NC $\downarrow$ | Max NC $\downarrow$ | NSC $\uparrow$ |
|---|---|---|---|---|---|---|---|---|---|
| 0.25 | 0.93 | 1.15 | 4 | 88 | 0.025 | 0.94 | 1 | 1 | 100 |
| 0.50 | 0.93 | 1.02 | 2 | 98 | 0.050 | 0.93 | 1.02 | 2 | 98 |
| 0.75 | 0.95 | 1.01 | 2 | 99 | 0.075 | 0.92 | 1.03 | 2 | 97 |

| $p_{\min}$ | Val $\uparrow$ | Avg NC $\downarrow$ | Max NC $\downarrow$ | NSC $\uparrow$ | $p_{\max}$ | Val $\uparrow$ | Avg NC $\downarrow$ | Max NC $\downarrow$ | NSC $\uparrow$ |
|---|---|---|---|---|---|---|---|---|---|
| 0.2 | 0.93 | 1.02 | 2 | 98 | 0.5 | 0.86 | 1.02 | 2 | 98 |
| 0.4 | 0.91 | 1 | 1 | 100 | 0.1 | 0.93 | 1.02 | 2 | 98 |
| 0.6 | 0.96 | 1.02 | 3 | 99 | 0.01 | 0.89 | 1.01 | 2 | 99 |
| 0.8 | 0.94 | 1.02 | 2 | 98 | 0.001 | 0.94 | 1.02 | 2 | 98 |

## D.3 Generated molecules

Figure 6 shows 6 (3 for the QED task, 3 for the LogP task) randomly selected molecules optimized by GRIDDD on the ZINC-250k dataset. For each molecule, we show 4 different successful optimizations. Figure 7 shows 30 random molecules generated without conditioning by GRIDDD on QM9.

## D.4 Failure Analysis

While analyzing the situations where GRIDDD fails in conditional generation the most, we have noticed how, unlike FreeGress, the experiments targeting $\mu$ tend to fail most frequently when targeting values close to zero. The few molecules generated tend to be small, split graphs. We have investigated the phenomenon and found out that the molecule with SMILES string CC (ethane) is known to have a dipole moment equal to zero. We believe that it is likely that GRIDDD tries to generate ethane or similar small molecules to minimize $\mu$, with higher failure rates than usual since these small molecules are under-represented in the dataset. Remarkably, DiGress and FreeGress are unable to pursue this minimization, since they almost inevitably start the reverse process from latents with a larger graph size and cannot adapt it to produce smaller molecules later on. Overall, this is an interesting behavior which shows that GRIDDD can successfully learn the properties of molecules with infrequent sizes as well.

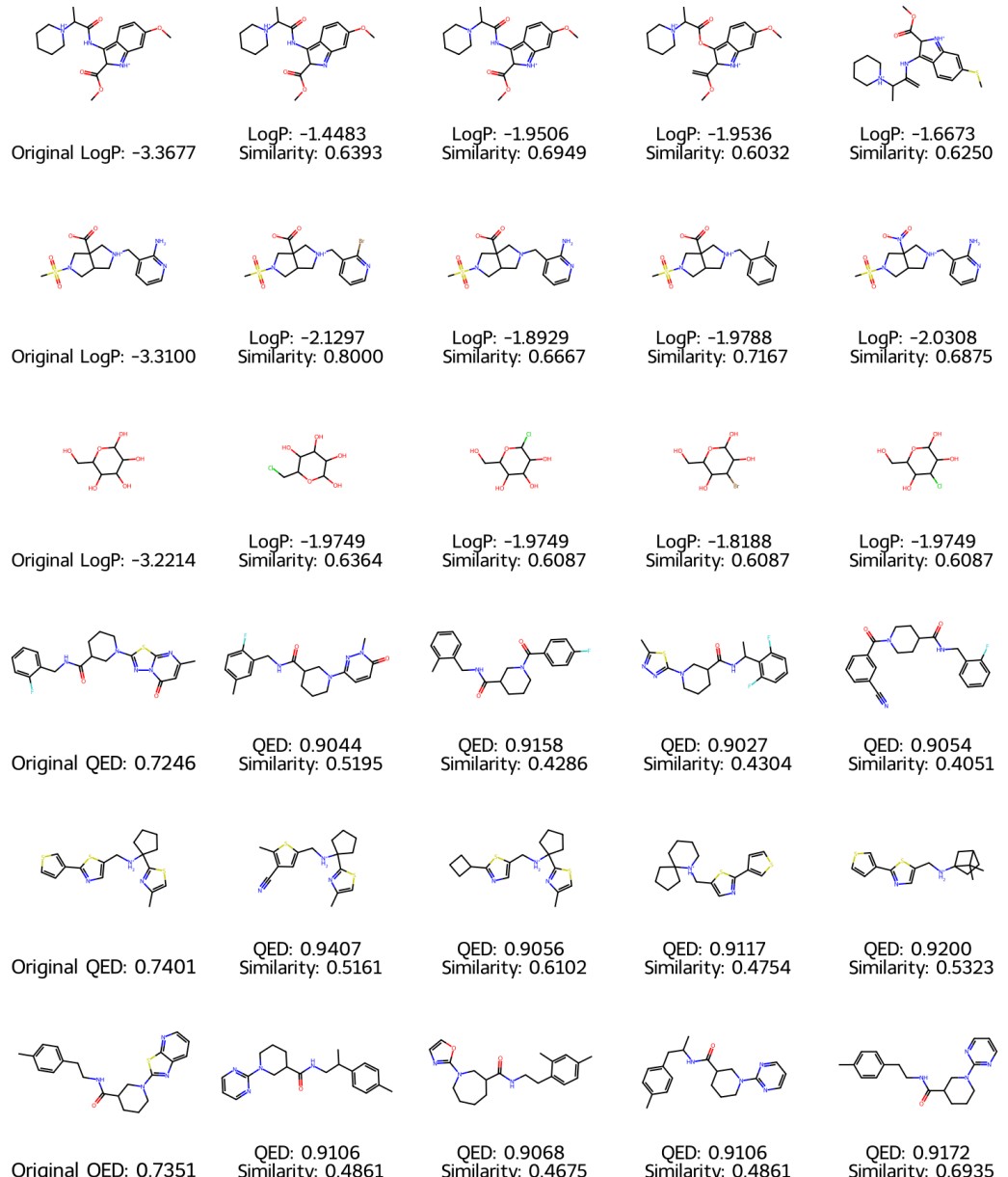

Figure 6: Randomly selected optimization experiments on ZINC-250k. The first element in each row is the original molecule, while the other four are results of different optimization experiments. We report, for each experiment, the original value and the resulting one.

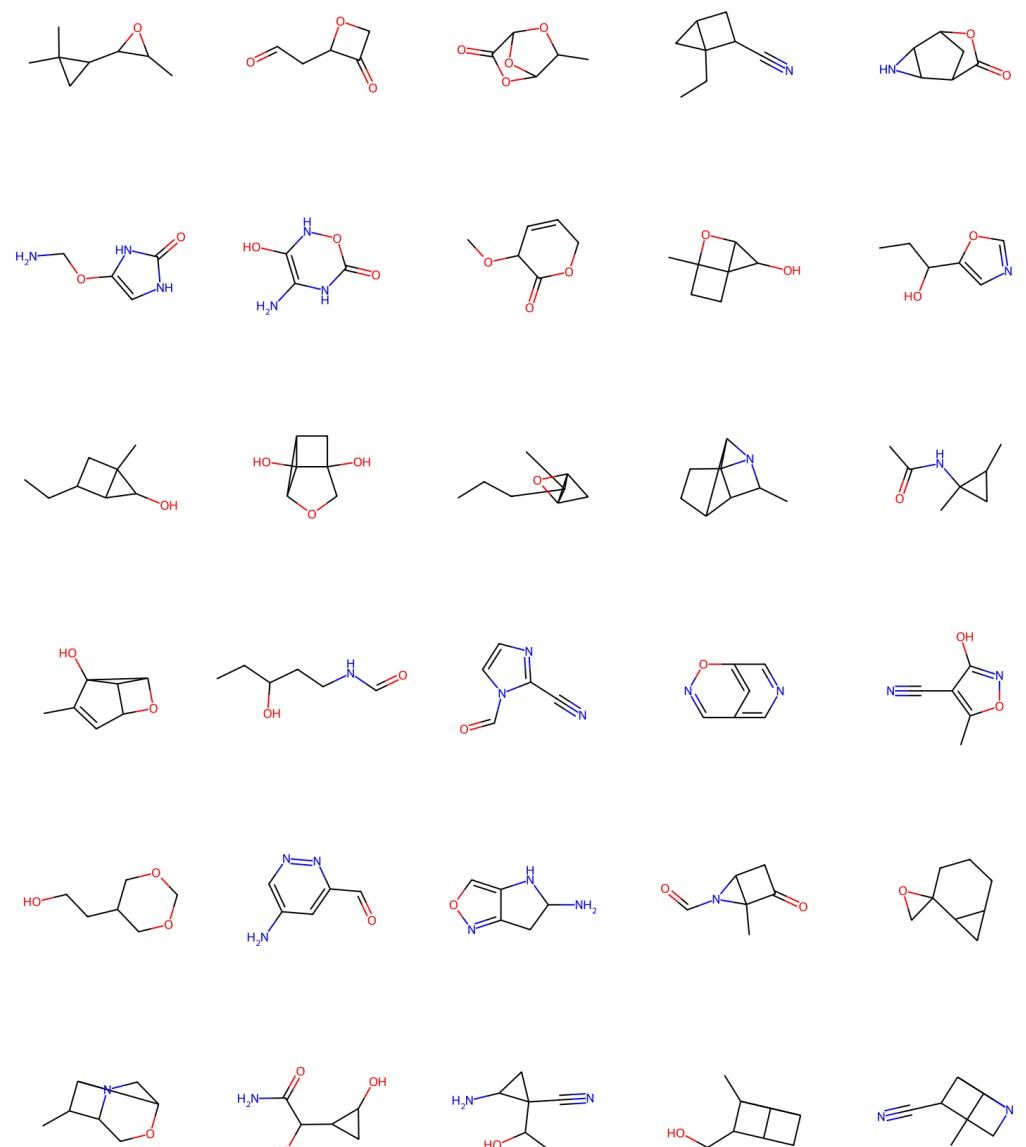

Figure 7: 30 non-curated samples generated without conditioning on QM9.

