# OpenReview forum: "Graph Diffusion that can Insert and Delete"
_NeurIPS.cc/2025/Conference — NeurIPS 2025 poster_

### Official Review · Reviewer_kZuT · 2025-07-01

**Clarity:** 2
**Significance:** 3
**Originality:** 3
**Rating:** 4
**Confidence:** 5

**Summary:**

The paper introduces GRIDDD, a novel discrete Denoising Diffusion Probabilistic Model (DDPM) for graph generation, which supports size-adaptive graph generation through monotonic insertions and deletions of nodes (atoms). Existing graph diffusion models have a fixed graph size throughout the generation process, limiting their application in tasks like property targeting and optimization, where graph size (number of atoms) is crucial. GRIDDD addresses this limitation by dynamically adjusting graph size during generation. It is evaluated on two tasks: property targeting (e.g., molecular weight, dipole moment) and property optimization (improving existing molecules). GRIDDD demonstrates superior performance on benchmark datasets (QM9 and ZINC-250k), particularly excelling in property targeting and optimization tasks.

**Questions:**

1. **Split Molecules Issue:** Could you elaborate on the causes behind split molecule generation? Have you considered additional constraints to mitigate this?

2. **Computational Overhead:** Have you explored optimization techniques (e.g., sparse updates, gradient checkpointing) to address computational overhead?

3. **Larger Graphs Generalization:** How might GRIDDD be adapted to effectively handle larger molecules beyond current limitations?

4. **Impact of Hyperparameters:** Could you provide sensitivity analyses regarding hyperparameters such as noise schedules and insertion/deletion probability distributions?

**Ethical Concerns:**

["NO or VERY MINOR ethics concerns only"]

**Final Justification:**

Based on the feedback, the three weaknesses I mentioned are indeed esixt and cannot be fully addressed at this stage. Therefore, although I consider this a good paper overall, I do not think it is appropriate to raise my score from weak accept to accept.

**Limitations:**

Yes

**Paper Formatting Concerns:**

No obvious formatting errors.

**Quality:**

3

**Strengths And Weaknesses:**

Strengths:

1. **Originality:** GRIDDD introduces a novel extension to DDPMs for graph generation by allowing node insertion and deletion during diffusion, significantly advancing beyond fixed-size models and enabling practical applications in molecular property targeting.

2. **Technical Contribution:** GRIDDD’s dynamic graph size adjustment during generation provides a substantial improvement in molecular generation tasks, effectively maintaining valid chemical structures throughout the process.

3. **Empirical Results:** The comprehensive experiments demonstrate GRIDDD’s superior performance on widely-used datasets (QM9 and ZINC-250k), achieving notable results in both property targeting and optimization, especially in enhancing LogP and QED scores.

4. **Clarity of Methodology:** The detailed technical explanations clearly illustrate how node insertions and deletions integrate into the diffusion process, including transition matrices and insertion-time predictions.

Weaknesses:

1. **Computational Overhead:** GRIDDD's insertion and deletion operations introduce significant computational overhead. The paper lacks concrete optimizations (e.g., sparse graph updates, mixed-precision training) or quantification of additional computational costs (training time, memory usage, FLOPs).

2. **Split Molecules:** GRIDDD occasionally generates more split molecules due to late-stage node insertions. Further exploration to address this issue through additional constraints or algorithmic modifications is needed.

3. **Limited Generalization to Larger Graphs:** GRIDDD’s effectiveness decreases significantly when generating molecules larger than its training distribution. Enhanced methods for improving generalization capabilities to larger graph sizes would be valuable.

---

> ### Author Rebuttal · Authors · 2025-07-30
>
> We thank the reviewer for the insightful review. Below, we respond to the reviewer’s concerns in order.
>
> - Computational Overhead: GRIDDD's insertion and deletion operations introduce significant computational overhead. The paper lacks concrete optimizations (e.g., sparse graph updates, mixed-precision training) or quantification of additional computational costs (training time, memory usage, FLOPs). Have you explored optimization techniques (e.g., sparse updates, gradient checkpointing) to address computational overhead?
>
> We have discussed the model's training time in the appendix (Section C.4), where we estimate that GrIDDD to be approx. 33% slower to train with respect to DiGress, denoting a typical tradeoff between expressivity/flexibility and computational overhead. Interestingly enough, we also observed that GrIDDD can, under some specific conditions, be more computationally efficient to sample than the alternatives. In particular, since our model is capable of starting the sampling process from a latent featuring any number of nodes, the reverse process can be much faster than DIGRESS if the initial latent is very small. As an example, GrIDDD takes approx. 42 seconds to sample 128 molecules on ZINC250k starting from latents with 2 nodes each, and approx. 64 when starting with 24 nodes (the most frequent size in the dataset). In contrast, DIGRESS, which needs to be initialized with a node matrix of size close to 36 to accommodate for the largest graph size in the dataset, requires approx. 75 seconds to run the same sampling task.
> As regards the number of parameters, GrIDDD uses a slightly larger but still comparable number with respect to FREEGRESS and DIGRESS (5M vs. 4.6M on QM9, 15.3 vs. 13.5 on ZINC250k). Overall, the tradeoff could be considered positive, since GrIDDD is more flexible than FREGRESS and DIGRESS and can be used for molecular optimization tasks. We commit to complement Section C.4 of the revised paper with these numbers and observations.
> As suggested by the reviewer, the slightly superior computational burden of GrIDDD could be mitigated by using appropriate optimization techniques. These would ultimately affirm GrIDDD as an comparable alternative to DIGRESS and FREEGRESS even in lower resource scenarios. However, we did not specifically employ any at this stage of development.
>
> - Split Molecules: GRIDDD occasionally generates more split molecules due to late-stage node insertions. Further exploration to address this issue through additional constraints or algorithmic modifications is needed. Could you elaborate on the causes behind split molecule generation? Have you considered additional constraints to mitigate this?
>
> Earlier in the development of GrIDDD, we analyzed where, in the reverse process, split molecules appeared, plotting the number of occurrences as a function of the denoising timestep. Overall, the statistics perfectly matched the shape of the $\zeta’(t)$ function which modulates the deletion schedule. As a result, we conjecture that split molecules are mostly caused by nodes inserted too late during the reverse process, which cannot be re-attached in time to the main graph, as the noise scheduler does not allow for too many changes in the graph’s structure during the last phases of the process.
> As regards implementing additional constraints, during our preliminary tests, we have found that GrIDDD generates less split molecules when given, as extra features, different powers of the adjacency matrix. This can be interpreted as a sufficiently accurate approximation of the number of connected components in the graph (as opposed to the computationally demanding task of computing the eigenvalues of the adjacency matrix's Laplacian). However, we have ultimately decided to proceed without this feature to ensure a fair comparison with the baselines. We will clarify these additional aspects regarding the issue of split molecules in the revised version of the paper.
>
> - Limited Generalization to Larger Graphs: GRIDDD’s effectiveness decreases significantly when generating molecules larger than its training distribution. Enhanced methods for improving generalization capabilities to larger graph sizes would be valuable. How might GRIDDD be adapted to effectively handle larger molecules beyond current limitations?
>
> First of all, we would like to state that according to our results, GrIDDD already takes a convincing step forward towards a more reliable out-of-distribution sampling with respect to existing approaches, as shown in Section 5.2 of the paper (which shows an improved capacity of generating valid molecules out-of-distribution with respect to the baselines). Clearly, we acknowledge that the problem is not completely solved and more research is needed, as generative models are trained to imitate the dataset distribution and not something they did not see during training. One possible direction to take in this regard is to parameterize both insertions and deletions (i.e., letting the neural network decide which kind of node to add, rather than relying on sampling from the marginal distribution of the nodetypes). In this sense, GrIDDD presents excellent opportunities to be extended and improved, and we plan to address the out-of-distribution sampling problem with innovative approaches in future studies. We will improve the conclusions section with this discussion.
>
> - Impact of Hyperparameters: Could you provide sensitivity analyses regarding hyperparameters such as noise schedules and insertion/deletion probability distributions?
>
> We have performed some experiments, tweaking the values of $p_{min}$ (we observed how $p_{max}$ can be kept equal to one, and that editing $p_{min}$ is enough), $D$, $w$, and the $\lambda$ associated with the insertion time's loss. We have trained several GrIDDD variants on QM9 while monitoring the behaviour of the training and validation losses, sampling 100 molecules from the trained models. The results are summarized in the table below. Overall, we observed that setting $D$ too small (which corresponds to starting the operations of insertion and deletion too early in the forward process) causes the model to generate higher rates of split molecules, since the model has little time to connect them back into the main graph while denoising. For values of $D$ higher than 0.5, we did not observe meaningful differences. We observed a similar behavior when setting large $w$ values, which was expected since this makes the "slope" of the $\zeta$ function less steep and, as a result, allows late reinsertions during the reverse process. Finally, we find the model to be relatively robust to different values of $p_{min}$, as well as the $\lambda$ associated with the insertion time loss (unless it is too high).
>
> - val: validity (higher=better)
> - avg. nc: average number of isolated connected components (lower=better)
> - nc max: maximum number of isolated connected components (lower=better)
> - n conn: number of valid connected graphs generated (higher=better)
>
> $$
> \begin{array}{ccccc}
> D & val & \text{avg. nc} & \text{nc max} & \text{n conn} \\\\
> \\hline
> 0.25 & 0.93 & 1.15 & 4 & 88 \\\\
> 0.50 & 0.93 & 1.02 & 2 & 98 \\\\
> 0.75 & 0.95 & 1.01 & 2 & 99 \\\\
> \end{array}
> $$
>
> $$
> \begin{array}{ccccc}
> w & val & \text{avg. nc} & \text{nc max} & \text{n conn} \\\\
> \\hline
> 0.025 & 0.94 & 1 & 1 & 100 \\\\
> 0.050 & 0.93 & 1.02 & 2 & 98 \\\\
> 0.075 & 0.92 & 1.03 & 2 & 97 \\\\
> \end{array}
> $$
>
> $$
> \begin{array}{ccccc}
> p_{min} & val & \text{avg. nc} & \text{nc max} & \text{n conn} \\\\
> \\hline
> 0.2 & 0.93 & 1.02 & 2 & 98 \\\\
> 0.4 & 0.91 & 1 & 1 & 100 \\\\
> 0.6 & 0.96 & 1.02 & 3 & 99 \\\\
> 0.8 & 0.94 & 1.02 & 2 & 98 \\\\
> \end{array}
> $$
>
> $$
> \begin{array}{ccccc}
> lamb & val & \text{avg. nc} & \text{nc max} & \text{n conn} \\\\
> \\hline
> 0.5 & 0.86 & 1.02 & 2 & 98 \\\\
> 0.1 & 0.93 & 1.02 & 2 & 98 \\\\
> 0.01 & 0.89 & 1.01 & 2 & 99 \\\\
> 0.001 & 0.94 & 1.02 & 2 & 98 \\\\
> \end{array}
> $$

---

> > ### Comment · Reviewer_kZuT · 2025-08-05
> >
> > While I recognize the strengths of this work, I will maintain my current score because the noted weaknesses are still relevant and require attention.

---

> > > ### Author Response · Authors · 2025-08-05
> > > **Thank you**
> > >
> > > We sincerely thank the reviewer for the overall positive reception of our paper, and in particular for recognizing its strengths. Even though we honestly thought our rebuttal had at least partially addressed the reviewer’s concerns, we witness that the judgement is somewhat definitive. Nonetheless, we are still seriously committed to improving its quality, also in consideration of the extended author-reviewers discussion period. As such, we are available to further discuss and clarify those parts of our work that the reviewer thinks remained relevant or in need of attention after our rebuttal.

---

> > > ### Comment · Area_Chair_yYza · 2025-08-05
> > >
> > > This response is insufficient. Please provide more details.

---

> > > > ### Comment · Reviewer_kZuT · 2025-08-05
> > > >
> > > > Based on the feedback, the three weaknesses I mentioned are indeed esixt and cannot be fully addressed at this stage. Therefore, although I consider this a good paper overall, I do not think it is appropriate to raise my score from weak accept to accept.

---

### Official Review · Reviewer_ckHS · 2025-07-02

**Clarity:** 3
**Significance:** 3
**Originality:** 3
**Rating:** 4
**Confidence:** 2

**Summary:**

GRIDDD extends discrete DDPMs for graphs by introducing mechanisms for node insertion and deletion within the forward (noising) and reverse (denoising) diffusion processes. It enables step-wise monotonic node insertions and removals during diffusion, and is specifically designed to change the graph size dynamically throughout the generative process, providing the necessary flexibility to incorporate conditional information more effectively.

**Questions:**

Failure Analysis: For cases where GRIDDD shows slightly lower validity (e.g., QM9, μ target), what are the common types of invalid molecules generated? Understanding these failure modes could guide future improvements.

**Ethical Concerns:**

["NO or VERY MINOR ethics concerns only"]

**Final Justification:**

Thank you for the detailed rebuttal and additional analyses. The clarification on the mutual independence assumption is helpful and addresses my concern.

The failure analysis for QM9 μ≈0 is insightful, and I appreciate the concrete example and explanation of how graph size adaptation may lead to higher invalid rates in rare-size molecules. This adds useful context to interpret the slightly lower validity in those cases.

The sensitivity experiments on pmin, pmax, D, w, and λ provide practical guidance for tuning on new datasets and indicate reasonable robustness in typical ranges, which is valuable for practitioners.

Overall, the rebuttal addresses my questions well, improves the clarity and completeness of the paper, and does not raise new concerns. I will maintain my borderline accept rating, as the paper is technically solid with meaningful contributions, though there is still some room to further discuss practical tuning guidelines in the final version.

**Limitations:**

Sensitivity to Hyperparameters: How sensitive is GRIDDD's performance to the choice of hyperparameters, particularly pmin, pmax, D, w, and the λ weights in the loss function? Are there general guidelines for tuning these for new datasets?

**Paper Formatting Concerns:**

Formatting is good.

**Quality:**

3

**Strengths And Weaknesses:**

Pros
Addresses a Critical Limitation: Directly tackles the inability of prior graph DDPMs to change graph size, which is vital for many real-world molecular design tasks (e.g., property-driven design where size correlates with properties).

Improved Performance in Property Targeting: GRIDDD matches or exceeds the performance of state-of-the-art graph diffusion models (DiGress, FreeGress) on molecular property targeting benchmarks (QM9 and ZINC-250k) in terms of Mean Absolute Error (MAE) and chemical validity.

Cons
Assumption of Mutual Independence: The reverse process factorizes probabilities assuming mutual independence of nodes and edges, which might be a simplification given the highly interconnected nature of molecules.

---

> ### Author Rebuttal · Authors · 2025-07-30
>
> We thank the reviewer for the insightful review. Below, we respond to the reviewer’s concerns in order.
>
> - Assumption of Mutual Independence: The reverse process factorizes probabilities assuming mutual independence of nodes and edges, which might be a simplification given the highly interconnected nature of molecules.
>
> The reviewer is correct in principle. However, all discrete graph diffusion models make the same assumption of mutual independence. As far as we know, there are no discrete diffusion models for graphs that do not make such assumption, as it significantly simplifies the underlying math. At the same time, we point out that the neural network tasked to compute the posterior $q(x_i^0|G^t)$ does not make use of that assumption, as it receives the entire graph in input to predict the categories of every node (and edge) in the graph. To summarize, we agree with the reviewer that this is an important limitation, although it is shared by all known discrete denoising diffusion models for graphs. We will extend the limitations section accordingly.
>
> - Failure Analysis: For cases where GRIDDD shows slightly lower validity (e.g., QM9,  $\mu$ target), what are the common types of invalid molecules generated? Understanding these failure modes could guide future improvements.
>
> One very interesting insight we have gathered is that, unlike FreeGress, the experiments targeting $\mu$ tend to fail most frequently when targeting μs close to zero. The few molecules generated tend to be small, split graphs. We have investigated the phenomenon and found out that the molecule with SMILES string "CC" (ethane) is known to have a dipole moment equal to zero. We hypothesize that GrIDDD tries to generate ethane or similar small molecules to minimize the dipole moment, with higher failure rates than usual since these small molecules are under-represented in the data. Remarkably, DiGress and FreeGress are unable to pursue this minimization, since they almost inevitably start the reverse process from latents with a larger graph size and cannot adapt it to produce smaller molecules later on. Overall, this is an interesting behaviour which shows that GrIDDD successfully can learn the properties of molecules with infrequent sizes as well. We will extend the revised version to include a discussion of a failure analysis.
>
> - Sensitivity to Hyperparameters: How sensitive is GRIDDD's performance to the choice of hyperparameters, particularly $p_{min}$, $p_{max}$, $D$, $w$, and the $\lambda$ weights in the loss function? Are there general guidelines for tuning these for new datasets?
>
> We have performed some experiments, tweaking the values of $p_{min}$ (we observed how $p_{max}$ can be kept equal to one, and that editing $p_{min}$ is enough), $D$, $w$, and the $\lambda$ associated with the insertion time's loss. We have trained several GrIDDD variants on QM9 while monitoring the behaviour of the training and validation losses, sampling  100 molecules from the trained models. The results are summarized in the table below. Overall, we observed that setting $D$ too small (which corresponds to starting the operations of insertion and deletion too early in the forward process) causes the model to generate higher rates of split molecules, since the model has little time to connect them back into the main graph while denoising. For values of $D$ higher than 0.5, we did not observe meaningful differences. We observed a similar behavior when setting large $w$ values, which was expected since this makes the "slope" of the $\zeta$ function less steep and, as a result, allows late reinsertions during the reverse process. Finally, we find the model to be relatively robust to different values of $p_{min}$, as well as the $\lambda$ associated with the insertion time loss (unless it is too high).
>
> - val: validity (higher=better)
> - avg. nc: average number of isolated connected components (lower=better)
> - nc max: maximum number of isolated connected components (lower=better)
> - n conn: number of valid connected graphs generated (higher=better)
>
> $$
> \begin{array}{ccccc}
> D & val & \text{avg. nc} & \text{nc max} & \text{n conn} \\\\
> \\hline
> 0.25 & 0.93 & 1.15 & 4 & 88 \\\\
> 0.50 & 0.93 & 1.02 & 2 & 98 \\\\
> 0.75 & 0.95 & 1.01 & 2 & 99 \\\\
> \end{array}
> $$
>
> $$
> \begin{array}{ccccc}
> w & val & \text{avg. nc} & \text{nc max} & \text{n conn} \\\\
> \\hline
> 0.025 & 0.94 & 1 & 1 & 100 \\\\
> 0.050 & 0.93 & 1.02 & 2 & 98 \\\\
> 0.075 & 0.92 & 1.03 & 2 & 97 \\\\
> \end{array}
> $$
>
> $$
> \begin{array}{ccccc}
> p_{min} & val & \text{avg. nc} & \text{nc max} & \text{n conn} \\\\
> \\hline
> 0.2 & 0.93 & 1.02 & 2 & 98 \\\\
> 0.4 & 0.91 & 1 & 1 & 100 \\\\
> 0.6 & 0.96 & 1.02 & 3 & 99 \\\\
> 0.8 & 0.94 & 1.02 & 2 & 98 \\\\
> \end{array}
> $$
>
> $$
> \begin{array}{ccccc}
> lamb & val & \text{avg. nc} & \text{nc max} & \text{n conn} \\\\
> \\hline
> 0.5 & 0.86 & 1.02 & 2 & 98 \\\\
> 0.1 & 0.93 & 1.02 & 2 & 98 \\\\
> 0.01 & 0.89 & 1.01 & 2 & 99 \\\\
> 0.001 & 0.94 & 1.02 & 2 & 98 \\\\
> \end{array}
> $$

---

> > ### Comment · Area_Chair_yYza · 2025-08-04
> >
> > Please engage in the discussion with the authors. The discussion period will end in a few days.

---

> ### Author Response · Authors · 2025-08-08
> **Dear reviewer**
>
> Greetings,
>
> As we are on the last day before the end of the discussion period, we kindly request the reviewer to provide us with feedback on our reply.
>
> We extend our gratitude for understanding.
>
> Regards

---

### Official Review · Reviewer_uVPd · 2025-07-03

**Clarity:** 3
**Significance:** 2
**Originality:** 2
**Rating:** 4
**Confidence:** 4

**Summary:**

The paper proposes a method that allows to vary the number of nodes in graph discrete diffusion models. In particular, this method allows to insert and delete nodes throughout both the forward and reverse processes. This is relevant for problems such as property optimisation, where the objective is to modify a molecule, parsed to a graph, such that given properties of the molecule are optimised, while still remaining structurally close to the initial molecule. The newly introduced diffusion process is combined with pre-existing classifier-free conditional guidance to target the tasks of property targeting, property optimisation, and out-of-distribution sampling.

**Questions:**

1. The removal of the edges connected to deleted nodes causes discontinuities in the edge diffusion processes as well. Did the authors analysed this component? Any relevant observation?
2. "One problem of using a single deletion type is that nodes will loop in the absorbing state indefinitely once taken the DEL type, while the reverse process is supposed to revert all deletions into normal atom types" (l. 150-152). This is not true for the marginal noise model employed in the diffusion process as, contrarily to masking discrete diffusion processes, it allows transition between every state at any timestep. Therefore, it seems to me that the motivation for the definition of the state Del* is incorrect. While I still see the importance of using the Del* state to signal to the model at sampling time to also predict for that node, the motivation of such formulation has to be modified. Please clarify.
3. The node and edge classes assigned at insertion time by sampling from the marginal distribution of nodes and edges, respectively, can be further motivated by the fact that the usual marginal noise model preserves marginal distributions across the forward process. Without this motivation, seems an ad-hoc trick. Did you try anything different?
4. An important baseline to incorporate consists of a DiGress implementation with, as similar to edges, a "non-existing" node class (equivalent to DEL) and let the process unroll. Edges connected to non-existing nodes should be treated consistently with how edges connected to DEL/DEL* nodes are handled.
 This process also allows to vary the number of nodes throughout the diffusion process without incurring into extra computational costs neither the more intricate diffusion process formulation of this paper. A superior performance to this baseline would clearly fundament the utilisation of the proposed diffusion process, with the additional model.

**Ethical Concerns:**

["NO or VERY MINOR ethics concerns only"]

**Final Justification:**

**Final Justification**

Following the rebuttal and discussion, several of my initial concerns have been addressed:

- The discontinuity in the edge diffusion process was acknowledged, justified with precedent from prior work, and proposed as future work.
- The broader applicability of the method beyond property optimization was clarified and convincingly argued. A particularly important shift in my perspective came from recognizing its potential for property targeting, especially for size-dependent properties, not only in molecular domains but across general graph settings. The flexibility to dynamically adjust the number of nodes during denoising is a distinctive and valuable capability. The results in Tables 1 and 2 support this: while GrIDDD does not outperform baselines when property–size dependence is minimal, it clearly excels when this dependency is stronger. The authors were encouraged to make this connection more explicit in the final version.
- Additional details will be included in the final version regarding: continuous diffusion models and dimension-varying approaches, a clear explanation of the sampling time overhead, improved formatting of Table 1 and Figure 3, justification of the newly introduced hyperparameters and their selection procedure, clearer motivation for sampling from the marginal distribution of nodes and edges, and comparison with DiGress baselines that include DEL / DEL* node and edge types.

While the motivation for the DEL* node type is now clearer, its necessity remains debatable (see discussion in rebuttal). It adds complexity and may limit potential benefits from remasking/self-correction.

**Weighting:** The clear articulation of broader utility and the supporting results significantly improved my assessment, outweighing remaining reservations. However, the extent of the score increase is limited by the amount of revisions still needed and the questionable necessity of the DEL* node type. My overall positive view is sustained by the fact that, to date, this is the only graph diffusion model that allows varying dimension (number of nodes).

**Limitations:**

Yes, the authors explicitly address the limitations of the paper and potential future directions.

**Paper Formatting Concerns:**

No concern.

**Quality:**

3

**Strengths And Weaknesses:**

**Strengths**:
1. The writing of the paper is clear.
2. The targeted problems (property targeting, property optimisation, out-of-distribution sampling) are relevant. The proposed method outperforms existing alternatives for property optimisation.
3. While the idea of varying the dimensionality of the generated instance by the diffusion model is not novel in general [1], neither for discrete state-spaces in particular [2], this work is the first, to the best of my knowledge, employing successfully such idea for graph diffusion models.
4. *Empirical results*: GrIDDD clearly outperforms existing alternatives for property optimisation setting. This method can also be used to ensure better out-of-distribution graph generation in terms of number of nodes, as its training allow the model to learn some representations over graphs that have more or less nodes than the ones in the training set (contrarily to vanilla discrete diffusion processes).
5. *Reproducibility*: the authors provide the code.

**Weaknesses**:
1. Empirical performance for property targeting is not remarkable for property targeting, as it only dominates existing methods (FreeGress) in 2 out of 5 different settings.
2. The paper does not compare/discuss to alternative graph continuous diffusion models (that model graph diffusion over continuous state-spaces), and their alternatives for varying dimensionality across diffusion process [1].
3. As mentioned by the authors in their limitations, this more intricate diffusion process incurs computational overhead. An objective analysis of this aspect of the contribution is of utmost importance for and correct evaluation of the contribution, e.g., to better understand the scalability of the method to larger molecular datasets.
4 While the utility of the method for property optimization is clear, I have some concerns regarding the pertinence of the proposed method for the other experimental settings.


**Minor weaknesses**:
1. The formatting of Table 1 and Figure 3 should be improved.
2. Still regarding reproducibility, it would benefit the paper to include a discussion on hyperparameter selection, particularly for the newly introduced hyperparameters. While the remaining ones follow prior work, a description of how the new parameters were chosen would strengthen the reproducibility and transparency of the experimental setup.


[1] - Trans-Dimensional Generative Modeling via Jump Diffusion Models, Campbell et al., NeurIPS 2023

[2] - Beyond In-Place Corruption: Insertion and Deletion in Denoising Probabilistic Models, Johnson et al., ICML 2021 Workshops

---

> ### Author Rebuttal · Authors · 2025-07-30
>
> We thank the reviewer for the insightful review. Below, we respond to the reviewer’s concerns in order.
>
> - Empirical performance for property targeting is not remarkable
>
> We strongly believe that our model specifically addresses a known limitation of all current discrete denoising diffusion models for graphs (including FREEGRESS) in a principled and elegant way. To our knowledge, this is the first method that can learn the graph size and adapt it to the generative task without resorting to estimates, heuristics, or auxiliary models. At the same time, it does so while performing on par or better than the existing baselines, meaning that its capability to reach and even surpass the same performance is actually a signal of better generalization capabilities than the former method. Moreover, our method is more widely applicable than  both FREEGRESS and DIGRESS (which lack the ability to adapt the graph size needed for optimization tasks).
>
> - The paper does not compare to graph continuous diffusion models
>
> We thank the reviewer for the reference. From our understanding, the main difference between jump diffusion and our solution is that the former can only reduce the number of entries in the data point during the forward process (in the case of molecules, these would be the atoms), while our model can also increase them. This implies that graph continuous diffusion cannot be adapted for property optimization tasks, since it lacks the capability of deleting additional nodes during the reverse process besides the ones deleted during the forward process. Overall, jump diffusion is better suited to be applied to continuous data such as videos and, more in general, to data with positional dependencies. Nonetheless, we will certainly add the reference to the revised paper and extend the related works section to discuss the differences between graph continuous diffusion models and GrIDDD.
>
> - An analysis of the computational overhead is required
>
> We have discussed the overall estimated overhead in the appendix, in section C.4. Clearly, it is unavoidable that a model that does more than a standard discrete diffusion will feature some sort of overhead. However, we take the opportunity to point out that GrIDDD can, under some specific conditions, be faster than the alternatives. In particular, since our model is capable of starting the sampling process from a latent featuring any number of nodes, the reverse process can be much faster than DIGRESS if the initial latent is very small. As an example, GrIDDD takes approx. 42 seconds to sample 128 molecules on ZINC250k starting from latents with 2 nodes each, and approx. 64 when starting with 24 nodes (the most frequent size in the dataset). In contrast, DIGRESS, which needs to be initialized with a node matrix of size close to 36 to accommodate for the largest graph size in the dataset, requires approx. 75 seconds to run the same sampling task.
>
> - Concerns regarding the pertinence of GrIDDD for property targeting
>
> We believe our solution is also well suited for property targeting. Clearly, the distinctive trait of GrIDDD is that it can flexibly adapt the graph size during the forward and reverse processes. Therefore, its best usage is to target properties that directly relate to graph size. However, in drug design, several properties (such as molecular weight, polar surface, and synthetic accessibility, and, to a minor extent, solubility) significantly correlate with the number of atoms. At the same time, although GrIDDD has been applied only on molecular datasets due to the abundance of available baselines and datasets, it can target or optimize arbitrary graphs, meaning that it can be employed for a variety of tasks beyond molecular generation.
>
> - Formatting of Table 1 and Figure 3
>
> We thank the reviewer for spotting these mistakes. We will fix them in the revised version of the paper.
>
> - Discussion on hyperparameter selection
>
> We thank the reviewer for the opportunity to clarify this point further. Overall, the choice of the hyperparameters was the result of a series of preliminary tests. We have chosen the values for $w$ as 0.05 and $D$ as 0.5 because they gave us some of the best results in terms of validity and number of connected components using a validation set. Setting $D$ any lower causes the model to generate many split molecules. The $\lambda$ associated with the insertion time of the nodes does not seem to affect training much, although setting it higher than 0.5 causes the model to underperform. Same thing for $p_{min}$  and $p_{max}$  (although it seems like setting $p_{min}$  between 0.2 and 0.4 bears slightly better results). We will add this discussion in the revised version of the paper to improve clarity and reproducibility.
> We believe this answer should resolve the reviewer’s concern; however, we remain open to provide the reviewer with more in-depth comparative results if needed.
>
> - Possible discontinuities in the edge diffusion processes. Any relevant observations?
>
> Removing edges during the forward process is not an issue, as each node and edge is corrupted independently without considering the rest of the graph. We have no relevant observations in this merit; we remain, however, curious to read whether the reviewer has any hypothesis in this sense that we could test.
>
> - Clarification on the DEL* node type (l. 150-152)
>
> We apologize for the lack of clarity, and we will correct the sentence in the final version of our paper. Specifically, we were referring to the reverse process. During the reverse step, in a setup with only one DEL type, a node with such a category can either switch to a "proper" class (the expected behavior) or remain a DEL. As a result, we defined the DEL* type, which is assigned to nodes marked for deletion. With this addition, we can guarantee that the nodes re-inserted during the reverse process will immediately switch back to a proper class, as the transition matrices do not allow for switching from DEL to DEL* during the forward process (and, hence, from DEL* to DEL while denoising, or DEL* to DEL* for that matter).
>
> - Motivations behind sampling from the marginal distribution of nodes and edges
>
> The reviewer is correct, this is the exact motivation behind this design choice. During our preliminary studies, we tried different options for completeness, such as always inserting the least frequent node/edge class. However, using the sample’s marginal distribution consistently gave us better performance, aligning to our expectations. We will update the revised text to include this motivation to better clarify our intent.
>
> - Comparison against DIGRESS baselines with DEL/DEL* node/edge types
>
> If we understand the reviewer’s intent correctly, we argue that it is not possible to treat the edges connected to non-existing nodes exactly as we treat the ones connected to DEL* nodes. In fact, GrIDDD removes edges altogether while the suggested variant has fixed graph size, and the edges would have to switch back and forth between DEL and “proper” types. As a result, DEL nodes could appear in the final latent $X^T$, and the associated edges would have to be set as DEL as well. This, in turn, would imply that the noisy graph $X^T$ is not sampled from an actual stationary distribution. GrIDDD solves this problem with a forward process that does not allow the presence of DEL nodes at the final corruption timestep $T$. Another minor issue would be caused by mismatches between the node and edge types (due to, for example, an edge to which non-DEL nodes are connected switching independently to DEL). A naive solution to these shortcomings would be to simply lift the constraint proposed by the reviewer everywhere but at timestep 0. By doing so, we can train the model to predict, at time step zero, that the edges associated to a DEL node have to be set to a specific category (be it a DEL or a “proper” edge class). We have run a comparison experiment to answer the reviewer’s inquiry, by implementing two baseline variants on QM9. One is essentially DIGRESS with an additional DEL atom type, while the other adds a further edge type EDEL to mark any bond whose endpoint is of type DEL.
> The results of the comparison are shown below.
>
> - val: validity (higher=better)
> - avg. nc: average number of isolated connected components (lower=better)
> - nc max: maximum number of isolated connected components (lower=better)
> - n conn: number of valid connected graphs generated (higher=better)
> - TV: total variation between generated distribution and the dataset’s (lower=better)
> - CE: Cross-Entropy (lower=better)
>
> $$
> \begin{array}{lcccccccc}
> Model & Val & Avg. NC & \\text{NC Max} & \\text{N Conn} & \\text{Val X CE} & \\text{Val E CE} & Atom TV & Edge TV \\\\
> \\hline
> GrIDDD & 0.97 & 1 & 1 & 100 & 0.44 & 0.32 & 0.016 & 0.01 \\\\
> DIGRESS+DEL & 1 & 1.56 & 8 & 69 & 0.47 & 0.37 & 0.047 & 0.10 \\\\
> DIGRESS+DEL+EDEL & 1 & 1.2 & 4 & 84 & 0.48 & 0.38 & 0.05 & 0.09 \\\
> \end{array}
> $$
>
> As can be seen, both baselines perform poorly when compared to GrIDDD, especially in terms of connected components, indicating that the model has difficulties in generating molecules with less than nine nodes. The validity rates appear higher, but a close inspection of the generated molecules shows that they are inflated by the presence of isolated carbon atoms which were not correctly attached to the main component.

---

> > ### Comment · Reviewer_uVPd · 2025-08-03
> > **Further clarifications**
> >
> > Thank you for your response, clarifications, and the additional experiments. The authors have addressed most of my concerns. However, I still have a few follow-up questions:
> >
> > **Discontinuities in the edge diffusion process:**
> > While the rebuttal states that nodes and edges are noised independently, the paper mentions: “If a node is set to category DEL or DEL*, so are the edges associated to it” (l. 168). To the best of my understanding, this coupling breaks the assumed independence between node and edge noising processes and introduces discontinuities in the edge noise process. Could the authors clarify why this is not the case? If it does create discontinuities, did you perform any analysis of its impact? Are there alternative approaches that could preserve smoother dynamics?
> >
> > **Clarification on the DEL\* node type:**
> > While I understand that introducing DEL* ensures that deleted nodes transition to a “proper” class in the corresponding step of the denoising process, it is still unclear why the model would not function correctly without it. As mentioned previously, the marginal model allows transitions between any pair of states at any time during the reverse process. This means that transitions between DEL and proper classes should be learnable without introducing additional complexity. Thus, the justification for introducing DEL* in the formulation still remains unclear to me.

---

> > > ### Author Response · Authors · 2025-08-04
> > > **on the discontinuity of the edge diffusion process**
> > >
> > > We are pleased that we have been able to address most of the reviewers' concerns. We will now address the question regarding the discontinuities in the edge diffusion process. The second concern is discussed in a separate comment due to lack of space.
> > >
> > > Indeed, in this respect the reviewer is correct: edge removals are not completely independent from node removals and this could create discontinuities in the edge noising process. However, in this choice, we conformed to previous approaches such as autoregressive diffusion [1], which applies a similar relaxation of independence by masking nodes together with the edges they are connected to. In this sense, we are not aware of alternative approaches avoiding such a discontinuity as hinted at by the reviewer: that itself would be a substantial body of work on itself, and we agree that this could be a very fascinating direction to explore in subsequent papers.
> > > That said, the effect of a node/edge deletion is local to the node being deleted and its associated edges, which are deleted together, and does not compromise the rest of the noising process. Also, notice that this relaxation does not affect the reverse process, as $p(e_{ij}^{t-1}|e_{ij}^{0}, e_{ij}^{t}, x_{i}^{t}, x_{j}^{t})$ is equal to $p(e_{ij}^{t-1}|e_{ij}^{0}, e_{ij}^{t})$. This stems from the fact that by design, $x_{i}^{t}$ or $x_{j}^{t}$ = DEL* only if $e_{ij}^{t}$ = DEL* as well (same for when $e_{ij}^{t}$ is equal to a proper class, which indicates that both $x_{i}^{t}$ and $x_{j}^{t}$ belong to a proper class as well), so conditioning on the nodes does not add useful information in practical scenarios. Lastly, the empirical results of GrIDDD suggest that this choice is not detrimental to performance.
> > >
> > > To summarize this discussion, we will clearly identify and justify the relaxation (and its consequences for the noising process) in the revised version, and we will add finding smoother alternatives as future work.
> > >
> > > [1] Kong et al. (2023). Autoregressive Diffusion Model for Graph Generation. Proceedings of the 40th International Conference on Machine Learning

---

> > > > ### Author Response · Authors · 2025-08-04
> > > > **on the role of DEL***
> > > >
> > > > The purpose of the DEL* type is to *guarantee* that a node set for deletion in the transition $(t-1) \to t$ of the forward phase gets denoised *exclusively* to a proper category in the mirroring transition $(t -1) \gets t$ of the reverse phase. By design, GrIDDD must re-insert a node at the same timestep it transitioned to a DEL state during the forward process. Such reinsertion implies switching from a DEL state to a proper category. However, the standard DEL type has an unconstrained transition dynamics (it can switch to any type, including DEL itself), therefore it is unsuited to accomplish the above goal. This results in a misalignment between the forward and reverse processes meaning that the step $(t-1) \to t$ (forward) would not have a single, deterministic counterpart in the step $(t-1) \gets t$ (reverse).
> > > >
> > > > One could argue that setting to zero the probability to stay a DEL at the next step would suffice, but we wanted to refrain from using such "tricks" and, instead, provide stronger guarantees by having two deletion states with clearly separated semantics.
> > > >
> > > > To further see why using a single DEL type does not provide the sought guarantee, let us craft a simplified example. Assume a simple setup with just two “proper” categories and only one DEL class. The $A$, $B$, $D$ matrices ($C$ is not needed, as it governs the transition to DEL*) shown in Figure 2 are now:
> > > > $$A = \begin{array}{ccc}
> > > > 1&0&0\\\\0&1&0\\\\0&0&1\\\\
> > > > \end{array}
> > > > $$
> > > > $$B = \begin{array}{ccc}
> > > > a&b&0\\\\a&b&0\\\\0&0&1\\\\
> > > > \end{array}
> > > > $$
> > > > $$ D = \begin{array}{ccc}
> > > > 0&0&1\\\\0&0&1\\\\0&0&1\\\\
> > > > \end{array}
> > > > $$
> > > > with $a+b = 1$. Suppose $x_t = DEL$. Its one-hot representation is $(0,0,1)$. We then compute the distribution $p(x^{t-1}|x^{s}, x^{t}) = \frac{x^{t}Q^t * \overline{Q}^s}{\overline{Q}^{t}{x^{t}}^{‘}}$. In this situation, $x^{t}Q^t$ and $\overline{Q}^{s}{x^{t}}^{‘}$ will be two vectors whose entries are potentially non-zero due to the $\alpha$ and $\zeta$ constants. Therefore, there is no guarantee that the last row and column of the resulting matrix will be equal to zero. In turn, once multiplied against $p_\theta(x^0|x^t)$, there is no guarantee that $p_\theta(x^{t-1} = DEL|x^t)$ will be zero.
> > > >
> > > > Conversely, introducing the DEL* state ensures that the final result of $x^{t}Q^t$ and $\overline{Q}^{s}{x^{t}}^{‘}$ will be vectors in the form $(k,l,0,0)$ (the last two columns being the elements associated to DEL and DEL*), with $k$ and $l$ being once again obtained by different products of the various noise and delete schedulers. Consequently, $p(x^{t-1}|x^{s}, x^{t})$ will be a matrix with the last two rows and columns (associated to DEL and DEL*) set to zero. This guarantees that multiplying $p(x^{t-1}|x^{s}, x^{t})$ by a generic $p_\theta(x^0|x^t)$ will always result in a vector $p_\theta(x^{t-1} |x^t)$ where the categories associated to DEL and DEL* are always equal to zero.
> > > >
> > > > To summarize, adding the DEL* state is a deliberate design choice to explicitly enforce a constraint on the transition states. We hope we have managed to clarify these points, but we nonetheless remain available for further clarification if needed.

---

> ### Comment · Area_Chair_yYza · 2025-08-04
>
> Please engage in the discussion with the authors. The discussion period will end in a few days.

---

> ### Comment · Reviewer_uVPd · 2025-08-05
>
> Thank you for your response.
>
> **Discontinuity of the edge diffusion process**: My concerns here are addressed. I believe the discussion included in the rebuttal, as well as the proposal to investigate smoother edge diffusion as future work, should be incorporated into the final version of the paper.
>
> **DEL\* node type**: My concern on this point still remains. The diffusion model should be, by design, capable of learning which nodes should remain in the DEL state at $t = 0 $. While I understand the motivation behind introducing the DEL\* token to enforce monotonicity in the reverse process, this design choice introduces two main limitations from my perspective: it complicates the formulation, and it restricts the potential for remasking/self-correction, which has been shown beneficial on masked diffusion models (as GrIDDD can be seen, with DEL as the Mask state). That said, I do not view either of these limitations as critical for the paper’s core message. In fact, they point toward promising directions for future work.
>
> As an additional outcome of the rebuttal process, I would also recommend that the final version of the paper includes: a discussion on continuous diffusion models and dimension-varying approaches, a clear explanation of the sampling time overhead, improved formatting of Table 1 and Figure 3, justification of the newly introduced hyperparameters and their selection procedure, clearer motivation for sampling from the marginal distribution of nodes and edges, and comparison with DiGress baselines that include DEL / DEL* node and edge types.
>
> Overall, I am positive about the paper and will raise my score. However, the extent of this improvement is limited by the number of revisions that are still necessary.
>
> A particularly important point that shifted positively my perspective was the discussion around the broader utility of the method beyond property optimization. I now see its potential for property targetting as well, especially for properties that depend on the size of the graph, not just in molecular domains but in graph settings more generally. The flexibility of dynamically adjusting the number of nodes during the denoising process is indeed valuable. The results in Tables 1 and 2 support this view: while GrIDDD does not outperform existing methods for tasks where property dependence on graph size is minimal, it excels in cases where that dependency is more direct. This intuition is hinted in the text, but I would encourage the authors to elaborate further and make it more explicit in the final version.

---

> > ### Author Response · Authors · 2025-08-05
> > **Thank you**
> >
> > We thank the reviewer for the time and effort in evaluating our paper. We are pleased that all the major concerns have been addressed, and that the discussion has clarified the contributions of GrIDDD, in particular regarding its effectiveness for molecular property targeting and optimization.
> >
> > We would also thank the reviewer for the opportunity to engage in this fruitful dialogue, which we believe has contributed to a clearer understanding of our work and its implications.
> >
> > We remain fully committed to fixing all formatting issues and improving the discussion of all the points mentioned by the reviewer in the main text. Due to space constraints, the ablation and the hyperparameter sensitivity studies will be documented in the appendix sections, but they will nonetheless be properly referred to in the main text for clarity and ease of access.

---

### Official Review · Reviewer_kSBz · 2025-07-17

**Clarity:** 2
**Significance:** 2
**Originality:** 2
**Rating:** 4
**Confidence:** 3

**Summary:**

The manuscript presents a method, GRIDDD, that extends the DIGRESS method for discrete graph diffusion by monotonic node insertion or removal. Original DIGRESS method shows excellent performance in molecular generation tasks, but number of nodes of the graph have to be specified beforehand. For some applications, e.g., drug design, one aims to generate a molecule with specific properties and the size of the molecule might not be known beforehand. The GRIDDD method allows to change the number of atoms during the diffusion process; in combination with classifier guidance this makes it possible to optimize the generated molecule towards a certain property. For this, the manuscript introduces new transition states for deleting nodes, a scheduling function that determines when and how likely a deletion/insertion happens as well as an auxiliary neural network that predicts how many nodes to delete at a specific time step.

**Questions:**

As stated above, I see the main advantage of this method in the theoretical possibility to insert nodes/edges at a very late timestep. Otherwise, I could just use the original DIGRESS method (either by inserting "DEL" nodes in the beginning, or by simply adding nodes in the middle of the diffusion process. But the latter will probably give me worse results as there is not enough time to properly denoise them). I do not really understand the experimental setup of the "property optimization" section. Do you start from a input molecule and only add a low/middle amount of noise to guarantee that the generated molecule is close to the input molecule? Or do you generate new molecules from random latent and just keep the ones that are close to the input molecule? I think an evaluation would be interesting where the input molecule is noised by a lower amount of timesteps (e.g. t=100) and then denoised with a classifier-guidance.

Second, as I stated above it's relatively trivial to implement node insertions/deletions in DIGRESS, although this naive implementation is probably not as good as a proper implementation. I would still be interested in how well such a baseline method performs in your evaluations.

Finally, I just do not understand why you need two different DEL states. Can you explain this in more detail please?

**Ethical Concerns:**

["NO or VERY MINOR ethics concerns only"]

**Final Justification:**

The underlying method feels very complicated, but the authors could demonstrate that the method outperforms simpler solutions such as adding DEL nodes in DIGRESS.

**Limitations:**

yes

**Paper Formatting Concerns:**

Nothing.

**Quality:**

1

**Strengths And Weaknesses:**

First of all, the limitation that graph size has to be specified beforehand is not as critical as it seems. One can simply start from multiple graph sizes and sample all of them, deciding for the best graph afterwards. This is more time consuming than changing graph size during denoising, though. Another possibility is to learn a predictor that estimates an optimal graph size for the given problem. This is done by FREEGRESS and the evaluation shows that the GRIDDD method performs more or less the same for most problems. Only for molecular weight, the GRIDDD method outperforms FREEGRESS clearly. However, this is a property that completely depends on graph size and might be somewhat arbitrary and, in practice, rather uninteresting.

I would argue that the most interesting problems for generative models are sampling of similar molecules that optimize certain properties (e.g., given an already known drug, find another one that is highly similar but has better water solubility). For such tasks the size of the optimized/generated molecule is often close to the input molecule anyways. The interesting question here would be rather: can I start from the input molecule and only add a little bit of noise to ensure that the generated molecule is close. Here, a method like GRIDDD could be advantageous, as it allows, in theory, to add new nodes rather late in the diffusion process. I'm not sure if the "Property optimization" section goes in this direction, as the description does not mention if they start from a random latent or from a input molecule.

The decision to use monotonic insertions and deletions make the underlying markov chain time irreversible. My feeling is that this might introduces a lot of problems and it also makes the underlying math very complicated. I wonder if there is any advantage in the way GRIDDD introduces insertions and deletions. The original DIGRESS model keeps the number of nodes fixed, but it already allows for arbitrary insertions and deletions of edges. The way it's done is that edges just have a special "delete" state which is treated similarly to their "single bond", "double bond" and so on states. During diffusion, edges can be inserted and deleted. The same would also be possible with atoms. One would still have to decide for a "maximum number of atom" constraint at the beginning, but besides that no changes of the method are necessary to allows node insertions/deletions. Clearly, there are a lot of disadvantages of such a method. For example, edges could connect deleted atoms. But it is already possible that more than 4 edges connect to a single carbon atom and the model just learns to prevent this. In my opinion the GRIDDD method should evaluate against this very simple baseline.

Finally, I found several small errors in the manuscript
- the negative sign in the probability density function zeta has to be removed, otherwise, all probabilities are negative
- the scaling parameter w has to be inverted, i.e. the manuscript states to use w=0.05 but they probably mean w=1/0.05. Otherwise, the exponential becomes extremely large and the zeta function very steep.
- equation 2 is missing an absolute, otherwise the probability is maximum when all atoms are deleted and minimum if all atoms are inserted. But the maximum should be at zero insertions/deletions.

---

> ### Author Rebuttal · Authors · 2025-07-30
>
> We thank the reviewer for the insightful review. Below, we respond to the reviewer’s concerns in order.
>
> - First of all, the limitation that graph size has to be specified beforehand is not as critical as it seems. One can simply start from multiple graph sizes and sample all of them, deciding for the best graph afterwards. This is more time consuming than changing graph size during denoising, though. Another possibility is to learn a predictor that estimates an optimal graph size for the given problem. This is done by FREEGRESS and the evaluation shows that the GRIDDD method performs more or less the same for most problems.
>
> The reviewer is correct to point out that existing methods resort to _pre-hoc_ (auxiliary models, estimation) or _post-hoc_ (multiple sampling) heuristics to accommodate for different graph sizes. In contrast, GrIDDD is an _ad-hoc_ strategy that directly learns the graph size according to the conditional generative task, while being as effective as, or better, than the baselines. This also brings advantages in terms of its broader applicability: in fact, while GrIDDD can be used for molecular optimization, FREEGRESS cannot (since it cannot adapt the graph size dynamically to optimize the property).
>
> - Only for molecular weight, the GRIDDD method outperforms FREEGRESS clearly. However, this is a property that completely depends on graph size and might be somewhat arbitrary and, in practice, rather uninteresting.
>
> We agree with the reviewer that molecular weight directly depends on graph size. We, however, respectfully disagree on the fact that it is not critical, somewhat arbitrary, and not interesting. A relevant number of molecular properties are directly relatable to molecular weight, and thus, to the graph size. For example, among the 5 Lipinski rules for drug development [1], the first one is indeed a constraint on molecular weight, and the others (number of rotatable bonds, number of hydrogen donors/acceptors, LogP to a minor extent) are related to the graph size. Other key properties that relate to graph size include polar surface or synthetic accessibility, which are crucial in drug design. Therefore, we argue that methods to adapt the graph size during the generative process are worth studying, researching, and improving.
>
> [1] Lipinski et al. (1997). Experimental and computational approaches to estimate solubility and permeability in drug discovery and development settings. Advanced Drug Delivery Reviews, 23(1–3), 3–25.
>
> - I would argue that the most interesting problems for generative models are sampling of similar molecules that optimize certain properties (...). For such tasks the size of the optimized/generated molecule is often close to the input molecule anyways.
>
> Again, the reviewer’s statement is correct. Nonetheless, the size of the molecule is likely to change during the optimization process (e.g., consider removing a toxicophore group from a compound to improve its safety, or adding carbon chains to improve hydrophobicity). Contrary to existing methods, GrIDDD is capable by design of adapting to different graph sizes without workarounds such as training larger adjacency matrices to accommodate the out-of-distribution graph sizes.
>
> - The interesting question here would be rather: can I start from the input molecule and only add a little bit of noise to ensure that the generated molecule is close. Here, a method like GRIDDD could be advantageous, as it allows, in theory, to add new nodes rather late in the diffusion process. I'm not sure if the "Property optimization" section goes in this direction, as the description does not mention if they start from a random latent or from a input molecule
>
> We thank the reviewer for spotting this shortcoming and for giving us the opportunity to clarify this point. We described this exact setup in Section 4.3 (Out-of-distribution sampling) in the paper, but we forgot to add it to the experimental setup of Section 4.2 (Property optimization) as well. Indeed, as guessed by the reviewer, for property optimization, we start from an input molecule, we corrupt it for $t=100$ timesteps, and then perform the denoising process as usual. The example suggested by the reviewer (adding new nodes late in the diffusion process) is exactly why GrIDDD is so advantageous and effective in the molecular optimization task.
>
> - The decision to use monotonic insertions and deletions make the underlying markov chain time irreversible. My feeling is that this might introduces a lot of problems and it also makes the underlying math very complicated.
>
> If the reviewer is referring to our use of absorbing states (DEL), there is already literature discussing similar strategies [2], which shows that diffusion models work regardless of whether the underlying Markov chain is irreversible or not. At the same time, we point out how our reverse process only deals with the reversible portion of the Markov chain, as it only deals with DEL* nodes which are not absorbing nodes (rather, a gateway between proper classes and the absorbing DEL state, but still reversible). The irreversible part of the process is dealt implicitly by just predicting when one or more DEL nodes should switch back to DEL*. Therefore, we do not consider this an issue or a limitation.
>
> [2] Austin et al. (2021). Structured Denoising Diffusion Models in Discrete State-Spaces. 35th Conference on Neural Information Processing Systems (NeurIPS 2021).
>
> - I wonder if there is any advantage in the way GRIDDD introduces insertions and deletions. The original DIGRESS model keeps the number of nodes fixed, but it already allows for arbitrary insertions and deletions of edges. (...) In my opinion the GRIDDD method should evaluate against this very simple baseline.
>
> As the reviewer already noticed, fixing the graph size to a large number of nodes to account for out-of-distribution graph sizes would be extremely resource-inefficient, with slower training and sampling times especially for datasets with larger molecules such as ZINC250k.
> With that being said, we have implemented two baselines as requested by the reviewer and tested them on QM9. One is essentially DIGRESS with an additional DEL atom type, while the other adds a further edge type EDEL to mark any bond whose endpoint is of type DEL.
> The results of the comparison are shown below.
>
> - val: validity (higher=better)
> - avg. nc: average number of isolated connected components (lower=better)
> - nc max: maximum number of isolated connected components (lower=better)
> - n conn: number of valid connected graphs generated (higher=better)
> - TV: total variation between generated distribution and the dataset’s (lower=better)
> - CE: Cross-Entropy (lower=better)
>
> $$
> \begin{array}{lcccccccc}
> Model & Val & Avg. NC & \\text{NC Max} & \\text{N Conn} & \\text{Val X CE} & \\text{Val E CE} & Atom TV & Edge TV \\\\
> \\hline
> GrIDDD & 0.97 & 1 & 1 & 100 & 0.44 & 0.32 & 0.016 & 0.01 \\\\
> DIGRESS+DEL & 1 & 1.56 & 8 & 69 & 0.47 & 0.37 & 0.047 & 0.10 \\\\
> DIGRESS+DEL+EDEL & 1 & 1.2 & 4 & 84 & 0.48 & 0.38 & 0.05 & 0.09 \\\
> \end{array}
> $$
>
> As can be seen, both baselines perform poorly when compared to GrIDDD, especially in terms of connected components, indicating that the model has difficulties in generating molecules with less than nine nodes. The validity rates of the baselines appear higher, but a close inspection of the generated molecules shows that they are inflated by the presence of isolated carbon atoms which were not correctly attached to the main component.
>
> - Finally, I found several small errors in the manuscript (...)
>
> We thank the reviewer for pointing the small errors out. We confirm that we used $w$=0.05. The slope is not too steep for our needs (unfortunately, we can not post pictures). Equation 1 is indeed the derivative (with respect to $t$) of $\zeta(t)=1/(1+exp(-(D - t) / w))$, the function used for the transition matrices, but we compute the absolute value of the gradient to turn $\zeta’(t)$ into a bell-shaped function (and then normalize). All errors will be revised whenever we will be allowed to edit the original submission.
>
> - (...) I do not really understand the experimental setup of the "property optimization" section
>
> We believe we have answered to this concern above. Nonetheless, we are open to further discussion or clarification, if needed.
>
> - (...) it's relatively trivial to implement node insertions/deletions in DIGRESS (...) I would still be interested in how well such a baseline method performs in your evaluations.
>
> We believe we have answered to this concern above. Nonetheless, we are open to further discussion or clarification, if needed.
>
> - Finally, I just do not understand why you need two different DEL states. Can you explain this in more detail please?
>
> The DEL* state is the first state that a deleted node enters during the forward process. Within the GrIDDD formulation, it is necessary since the model is taught to learn the exact timestep where a node gets deleted. This is because, when denoising the sample at step t, we force a newly inserted node to switch immediately back to a "proper" node category at step $t-1$. The most intuitive category to assign at re-insertion would be DEL. However, the semantics of the DEL state is two-fold: it may either turn back into a proper node category, or it could stay in the absorbing state DEL (in the reverse process). Therefore, we determined that the best course of action is to treat the first timestep after a forward deletion as its own class (which we called DEL*). By doing so, the reverse process will always switch to a "proper" class at step $t-1$. We hope that our answer resolved the reviewer’s doubts, and we remain open to discussing further clarifications.

---

> > ### Comment · Reviewer_kSBz · 2025-08-07
> >
> > I agree that many properties correlate with mass, but this does not mean mass is an interesting optimization property. For drug-likeness people are simply interested in small molecules, but not in molecules with a specific number of atoms or a specific mass. I still think that the problem of searching for a molecule with a specific mass is very artificial and not very useful in real-world.
> > I agree, however, that the ability of changing the graph size is very useful for molecule optimization. The evaluation against the baseline of DEL-nodes in DIGRESS is very relevant and should be part of the manuscript.
> > The method does not feel very elegant but rather very complicated compared to the original DIGRESS, but for some applications this additional complexity might be necessary.

---

> ### Author Response · Authors · 2025-08-08
> **Thank you**
>
> We thank the reviewer for the time and effort in evaluating our rebuttal. We are glad that we both agree on the utility of adapting the graph size for molecular optimization.
>
> **Comment on molecular weight (MW)**
>
> While we acknowledge that MW may not be the most suitable property to be targeted in drug design, in this paper it serves two specific purposes:
>
> - Since it strongly correlates with the number of nodes of the molecular graph, it provides a natural and intuitive testbed to assess the ability to flexibly adapt the graph size, which is precisely what GrIDDD was designed for.
> - It enables a fair and comprehensive comparison with FreeGress, which first introduced this target in its experimental setting. Note that FreeGress is our direct competitor, achieving state-of-the-art performance in property targeting while also including a mechanism to control the graph size.
>
> We commit to better explain the rationale behind choosing MW in the revised version.
>
> On a more general note, we would also like to emphasize that GrIDDD has been designed *to dynamically control the graph size*, and not to optimize any molecular property in particular. Therefore, our contribution is primarily methodological and intended for the broader task of targeted graph generation, although we were constrained to apply it in the molecular domain due to the abundance of baselines. From this broader perspective, we see GrIDDD as a step forward in enhancing the flexibility and generality of discrete diffusion models for graphs, opening the door to applications beyond drug design (e.g., [1,2]).
>
> **Comment on the complexity of the model**
>
> We agree that GrIDDD introduces additional components when compared to seminal models like DiGress. However, we believe these additions are minimal and principled. Aside from the additional scheduler, the only modifications involve two extra rows and columns in the $A$ and $B$ matrices, and two sparse matrices $C$ and $D$, with at most one nonzero entry per row. Techniques such as insertions and deletions have been widely studied [3], although not for graphs. The same applies for the matrices $\overline{Q}_{t|s}$ [4] and absorbing diffusion [5,6]. As the reviewer correctly noted, the ability to adapt the graph size can offer a clear advantage in terms of flexibility (since our model is the only one capable of adapting the graph size without rough workarounds) and effectiveness (as shown by the empirical results and the ablations). In this light, we view the added components not as a detour from elegance, but as a natural extension motivated by necessity and well-justified by the empirical improvements.
>
> **Comment on the baseline**
>
> We will include the experiments with the baselines in the main paper, and we thank the reviewer for pointing us to this useful ablation that has significantly strengthened our contribution.
>
> **Final request**
>
> We would like to ask whether our rebuttal and this discussion have sufficiently addressed the reviewer’s concerns. We remain open to further discussion if needed.
>
> ---
>
> References:
>
> [1] Cheng et al. (2024). Parallel Vertex Diffusion for Unified Visual Grounding. Proceedings of the 38th AAAI Conference on Artificial Intelligence.
>
> [2] Shabani et al. (2022). HouseDiffusion: Vector Floorplan Generation via a Diffusion Model with Discrete and Continuous Denoising. The IEEE/CVF Conference on Computer Vision and Pattern Recognition (CVPR).
>
> [3] Johnson et al. (2021). Beyond In-Place Corruption: Insertion and Deletion In Denoising Probabilistic Models. Workshop on Invertible Neural Networks, Normalizing Flows, and Explicit Likelihood Models (ICML 2021).
>
> [4] Zhao et al. (2024). Unify Discrete Denoising Diffusion. Preprint.
>
> [5] Austin et al. (2021). Structured Denoising Diffusion Models in Discrete State-Spaces. 35th Conference on Neural Information Processing Systems (NeurIPS 2021).
>
> [6] Kong et al. (2023). Autoregressive Diffusion Model for Graph Generation. Proceedings of the 40th International Conference on Machine Learning.

---

### Author Response · Authors · 2025-08-09
**Answer to all reviewers**

We sincerely thank all the reviewers for their valuable comments. We will make sure to include, in the final version of the manuscript, all the feedback received (some of it will have to be inserted in the appendix because of space constraints). While the full list of changes is too long to list here, it will include:

- A comparison of our model against a version of DiGress with DEL nodes and edges;
- A study on the influence of the new hyperparameters;
- Further clarifications on the role of the DEL* class;
- A discussion on the differences between our model and graph continuous diffusion models;
- A more in-depth discussion of the sampling time overhead;
- Clarifications on the molecular optimization procedure;
- Corrections to some formatting issues.

We will make sure to include the rest of the feedback we did not list here as well.

We hope we have solved all the major concerns during the Rebuttal and the Reviewer Author Discussions.

Regards

---

### Decision · Program_Chairs · 2025-09-17

**Decision:**

Accept (poster)

**Comment:**

This paper provides an important advance in graph diffusion models with a focus on molecule generation. In particular, the authors provide an approach that allows the number of nodes in the graph/molecule to vary, whereas previous work required the number of nodes to be predetermined. This is an extremely important property for many applications in molecule generation.

While some of the reviewers were a little dissatisfied with some of the specifics of the approach, the authors' method does work well at what it was designed to do with no current competitors. Based on my reading of the reviews and the responses, I believe that the authors addressed the reviewers' concerns well.